# Modeling Focal Synaptic Degeneration and Neural Plasticity in Ventral Visual Cortex

## Abstract

Strokes affect a significant portion of the population and often result in secondary damage in the form of focal synaptic degeneration. When this occurs in the ventral visual cortex (VVC), it can lead to neurological deficits, including visual function loss. In this paper, we use the VVC as a framework in which to model focal synaptic degeneration and post-injury plasticity. We do so by progressively "injuring" synaptic connections in primate visual areas V1, V2, V4, and the inferior temporal cortex (IT), followed by continual retraining of the spared connections on real-world visual stimuli. We demonstrate that the functional signatures of carefully designed differential tasks can localize synaptic decay in the VVC. Initially, categorization performance deteriorates gradually, up to a critical threshold, beyond which there is a sharp drop. This slow decline in performance is marked by a reorganization in nearby neurons, where both visual function and the structure of receptive fields adapt to compensate for the damage. Spared recurrent connections significantly contribute to recovery. Furthermore, we find that the presence of teaching signals in the form of category labels during rehabilitation leads to improved categorization performance recovery.

## 1 Introduction

Ischemic strokes, which account for approximately 62% of all incident strokes and are the third-leading cause of death and disability worldwide (Feigin et al., 2021), occur when an artery is blocked, leading to decreased blood flow, oxygen, and glucose in affected brain regions (Sacco et al., 2013; Janardhan & Qureshi, 2004). The posterior cerebral artery supplies the ventral visual cortex (VVC); thus, infarctions in this territory (∼10% of stroke cases) often result in focal visual deficits such as cortical blindness, achromatopsia, prosopagnosia, and pure alexia (Rubens & Benson, 1971; Milner et al., 1991; Hodges et al., 1995; Farah, 2004; Crutch et al., 2012; Lehmann et al., 2012; Maia da Silva et al., 2017; Martinaud et al., 2012; Robotham et al., 2023) (see Figure 1).

Neurological injury such as that due to synaptic activity degeneration, one of the earliest consequence of an ischemic stroke, is known to trigger mechanisms of post-injury plasticity in the somatosensory (Mogilner et al., 1993; Borsook et al., 1998), motor (Dancause et al., 2005; Kantak et al., 2012; Nudo, 2013), auditory (Collignon et al., 2009; Lomber et al., 2010), and visual cortices (Baker et al., 2005; Voss et al., 2017; Mikellidou et al., 2019). Plasticity in the form of functional reorganization and rewiring of intra-cortical synaptic connections has been observed after damage to the retina in the human visual cortex (Dilks et al., 2007), in adult cats (Schmid et al., 1996; Eysel & Schweigart, 1999), in mice (Keck et al., 2008), and in monkeys (Gilbert & Wiesel, 1992). In several cases, reorganization is brought about by a slight increase in the receptive field size of neurons at the borders of lesions (Kaas et al., 1990; Gilbert & Wiesel, 1992; Eysel et al., 1999; Papanikolaou et al., 2014).

While mechanisms of post-injury plasticity in the visual cortex can aid in compensating for the partial or complete loss of certain regions to some extent, visuoperceptual rehabilitation therapies like scaffolded training, becoming increasingly available, have a significant potential of helping with improving not just the patient's visual abilities, but also their everyday functioning and quality of life (Choi & Twamley, 2013; Heutink et al., 2019; Saionz et al., 2021).

Analyzing the interplay between lesion progression, time, synaptic dynamics, plasticity, and retraining protocols in silico using computational models can help to narrow down the search space for promising hypotheses around optimal recovery that would otherwise require significant time, money, and

Figure 1: **Focal degeneration in the ventral visual cortex.** Synaptic degeneration, secondary to focal ischemic strokes to different regions within the VVC, is part of an ischemic *cascade* that occurs post infarction. When a certain region is ischemically injured due to an occluded blood vessel (called the ischemic core), the surrounding "at risk" region undergoes markedly lowered tissue perfusion that is barely sufficient to support cellular function (called the ischemic penumbra). According to Moskowitz et al. (2010), the ischemic penumbra is potentially salvageable. However, quick action is required since, over time, the infarct core expands into the ischemic penumbra, reducing the chance for therapeutic intervention. Focal damage can lead to impaired performance on specific visual tasks such as face recognition, discriminating between color and contrast, category selectivity, etc. Because different regions are specialized to carry out different functions, and the cortex has mechanisms for recovery, in this paper we computationally investigate this dynamic interplay.

resources to be performed in animal or human subjects. Task-optimized, image-computable deep artificial neural networks (DANNs) are promising for generating hypotheses, as they are known to effectively predict responses in the healthy brain's VVC (Yamins et al., 2013; 2014; Khaligh-Razavi & Kriegeskorte, 2014; Majaj et al., 2015; Yamins & DiCarlo, 2016; Schrimpf et al., 2018; Nayebi et al., 2018; Kubilius et al., 2019; Cadena et al., 2019; Schrimpf et al., 2020; Zhuang et al., 2021; Finzi et al., 2022). Furthermore, they have been previously used to design and implement in a real animal brain (mice and macaques) optimal perturbations that produce stronger responses in neuronal sub-populations of the VVC than any previously known natural stimulus (Bashivan et al., 2019; Walker et al., 2019; Ponce et al., 2019). This is necessary for us, because when we are perturbing different areas of the model, as we have done in this work, we want the responses to be good enough that they can be optimized to drive the brain. Given that synaptic degeneration is a natural in-brain perturbation, DANNs seem that they are likely to be good at predicting those as well.

Prior works have extensively focused on modeling *global* damage to the VVC using purely feed-forward, fully supervised, artificial networks, often trying to understand what happens internally in the model compared to its healthy state and how this change relates to human or animal behavior (Hinton et al., 1993; Raj et al., 2012; Lusch et al., 2018; Tuladhar et al., 2021; Moore et al., 2021; 2022; 2023a;b). Through this paper, we are the first, to the best of our knowledge, to ask how model behavior changes under *focal* degeneration of different regions in the VVC in the presence of a post-injury plasticity mechanism. We use different classes of models—feedforward convolutional networks, those with intra-layer recurrence, those trained using self-supervision, etc.—to not only understand human behavior but also generate hypotheses about mechanisms and protocols supporting optimal recovery in the VVC. Specifically, we address the following:

1. It is well-known that the VVC is hierarchically organized (Felleman & Van Essen, 1991; Connor et al., 2007; Rust & DiCarlo, 2010; DiCarlo et al., 2012), with different regions specialized to perform different functions: V1 cells are orientation and spatial frequency selective (Carandini et al., 2005; Kamitani & Tong, 2005) while IT is selective to moderately complex object features (Tanaka, 1996) and faces (Kanwisher et al., 1997). Given such functional specialization, can we design a battery of visual tests that target these specific functions for localizing lesions within the VVC? (section 4.1)

2. In response to degeneration, to what extent can we recover object recognition performance because of neural plasticity? What is the scale of the number of visual stimuli needed for this to happen? And are there limits on the recovery capabilities of different regions when undergoing degeneration? (section 4.2)

3. What underlying mechanisms allow for recovery to occur? Do spared peri-lesional synapses surrounding the lesions compensate for the loss of its visual function, as is seen biologically? (section 4.3)

4. What roles do recurrent connections (Lamme & Roelfsema, 2000; Tang et al., 2018; Kar et al., 2019) and the type of task performed during the recovery phase play with respect to categorization performance recovery? (sections 4.4 and 4.5)

## 2 RELATED WORKS

There are numerous prior works that have studied pruning (i.e., an implementation of network degeneration) by introducing one-shot and iterative lesions to a DANN. Most of them do so to find sub-networks that offer better generalization, computational optimization, and storage efficiency, often inspired from biologically-implausible schemes like magnitude-based pruning, importance scores, particle-filters, pruning convolution filters, channels, and depth, and structured pruning (LeCun et al., 1989; Gorodkin et al., 1993; Hassibi et al., 1993; Han et al., 2016; Li et al., 2017; Molchanov et al., 2017; Yang et al., 2017; Zhu & Gupta, 2018; Anwar et al., 2017; He et al., 2018; Gao et al., 2019; Liu et al., 2019; Luo et al., 2018; Yu et al., 2018; Lin et al., 2019; Blalock et al., 2020; Meng et al., 2020; Wang et al., 2021; Yu et al., 2022). Quantitatively, the results that are explored in those works that perform selective pruning work with relatively small network sparsity levels. Models are often retrained on the entire training dataset for large numbers of epochs in a supervised fashion. In this work, we analyze network pruning for sparsity levels in the limit ($>99\%$), when models are retrained on various fractions of training images for constrained periods of time.

Outside of the literature on pruning as a way of computational optimization, one of the first papers to simulate brain damage was that of Hinton et al. (1993), where the authors use a shallow feedforward network to reproduce reading errors in deep dyslexia. Raj et al. (2012) mathematically model the diffusion of misfolded tau and beta amyloid to generate a predictive model of dementia. Lusch et al. (2018) analyze a series of global injury protocols applied to convolutional neural networks (CNNs)—randomly injuring $p$ percent of convolution and fully connected layer weights by setting them to zero; performing magnitude-based pruning; and using statistical data on Focal Axonal Swellings (FAS) to either block (set to 0), transmit (leave as is), reflect (divide by half), or filter (using a low-pass filter) weights—to show that the model makes more human-like mistakes.

More recently, Tuladhar et al. (2021); Moore et al. (2021; 2022; 2023a;b) have simulated global neurodegeneration and neural plasticity using CNNs by progressively injuring and retraining model weights and analyzing how representation dissimilarity matrices (RDM) and average Brain-Scores (Schrimpf et al., 2018; 2020) of lesioned, retrained, and healthy models compare with each other. A limitation of these works is that they only analyze how global ischemia and network retraining affect overall performance on object recognition, without trying to inspect the underlying mechanisms that bring about those results. Here, we distinguish ourselves by not just using a collection of biologically-plausible DANNs with and without recurrent connections and those trained using different objectives, but also being the first, to the best of our knowledge, in modeling focal ischemic strokes to different visual areas along the ventral pathway. We go beyond just observing what happens by asking ourselves why we see the model behavior that we see, to generate hypotheses around optimal recovery and rehabilitation strategies that can then be validated in vivo through clinical experiments.

## 3 METHODS

We divide our experiments and results into different sections based on the four guiding questions presented at the end of section 1. We use AlexNet (Krizhevsky et al., 2012), CORnet-Z (Kubilius et al., 2018), and ResNet-18 (He et al., 2016) as feedforward supervised model classes, CORnet-R (Kubilius et al., 2018) as a supervised model class with recurrent dynamics that propagate through the network in a biologically-valid manner, CORnet-S (Kubilius et al., 2019) as a supervised model class with skip-connections and within-area recurrence, and MobileNetV2 (Sandler et al., 2018) as a self-supervised model class trained using three different objectives—DINO (Caron et al., 2021), MoCo (He et al., 2020), and SwAV (Caron et al., 2020).

These models have been quantitatively shown to have a reasonably good performance on ImageNet (Deng et al., 2009), as well as be good predictive models of the VVC (as shown on BrainScore),

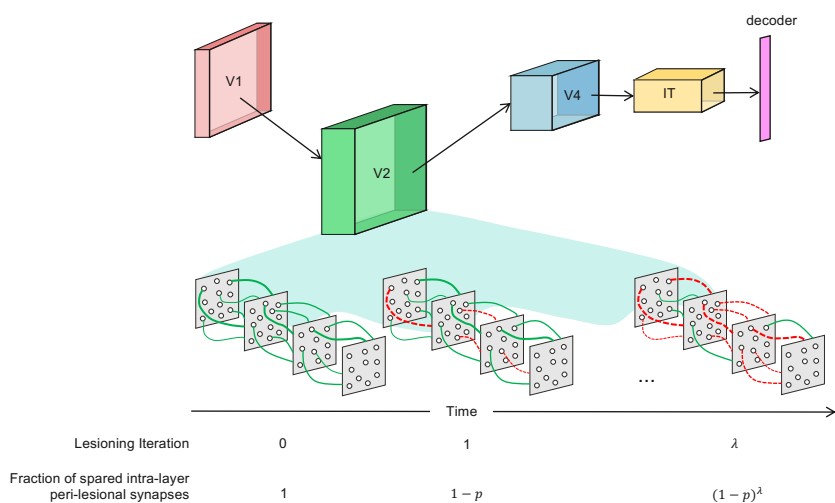

Figure 2: **Schematic of the degeneration mechanism.** For each region V1, V2, V4, and IT, we, across separate experiments, randomly injure $p$ fraction of the filter weights from convolutional layers in that region progressively for some $\lambda$ number of iterations. For example, if we are modeling degeneration in V2, and V2 has three convolutional layers, we will injure, say, $p = 0.2$ fraction of filter weights randomly from those three convolution layers every iteration for, say, $\lambda = 40$ iterations.

suggesting that their internals match the brain's anatomical and functional constraints. Furthermore, they have been shown to exhibit neuroanatomical consistency, in that early layers of the model are good predictive models of early visual cortex, intermediate layers are good predictors of V4, and higher layers are good predictors of neural responses in the IT. CORnet-S, specifically, is a good image-by-image human behavior predictor on categorization tasks. It is shallower than very deep models like ResNet-50, which are not anatomically consistent since there are somewhere on the order of 15-20 different visual areas in humans (Van Essen, 2003). Additionally, CORnet-S is a recurrent model, which makes it one of the few known models to produce accurate predictions of image-by-image temporal trajectories in IT neural responses over time (Kubilius et al., 2019).

### 3.1 MODELING FOCAL SYNAPTIC DEGENERATION

We first model focal synaptic degeneration in different visual areas of a network (section 4.1). Model layers are assigned to one of four visual areas—V1, V2, V4, and the inferior temporal (IT) cortex—by scoring their ability to predict real neuron responses in macaques (Freeman et al., 2013; Majaj et al., 2015), while maintaining neuro-anatomical consistency.

To introduce focal synaptic degeneration to one of the visual areas, we start with a model pre-trained on ImageNet (Deng et al., 2009) and then progressively and non-selectively prune filter weights from convolutional layers in that area (Hofmeijer & van Putten, 2012). The pruning scheme that we use is the same as that in Hinton et al. (1993). The pruning method follows a uniform schedule by masking a constant fraction $p$ of the healthy synaptic connections in the area at every lesioning iteration, for a total of $\lambda$ iterations (figure 2). Masking synaptic connections leaves them untrainable, implying necrosis, or cell death (see Lipp & Bonfanti (2016) for an evaluation of the variations and confusions around neurogenesis in adult mammals). This means that at any lesioning iteration $\ell \in [\lambda]$, the number of spared synapses will be given by the expression $1 - (1 - p)^\ell$. One might ask here if this is exactly how synaptic dysfunction occurs in the brain. We provide discussion on the same in section 5.

Given this mechanism, we analyze how degeneration in model V1, V2, V4, and IT affects object recognition performance on three different visual tasks with different functional signatures (figure 3A): *Labeled Faces in the Wild dataset* for face verification, *Contrast Sensitivity dataset* for contrast discrimination, and *Noisy Operators dataset* for shape detection under noise. A complete description of the tasks is presented in suppl. B.

All evaluations are conducted by training a linear probe on the pre-logits network features using the cross entropy objective on these datasets.

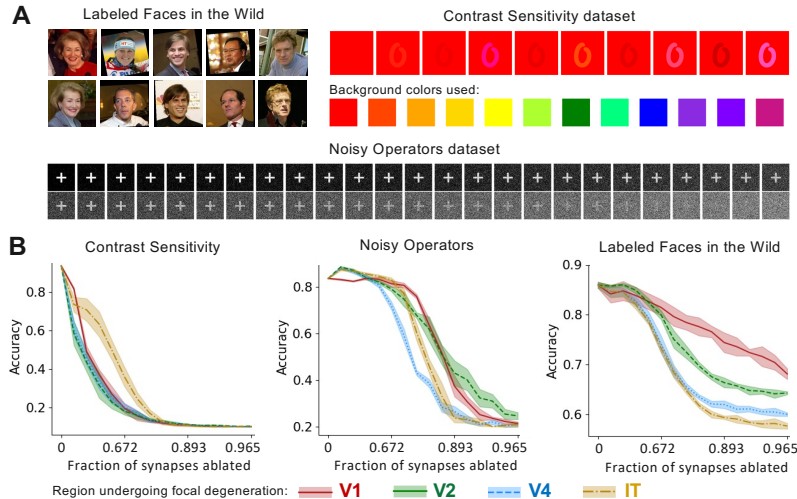

Figure 3: **Functional signatures of carefully designed differential tasks can help localize focal damage. A)** Visual tasks used to analyze how focal degeneration in CORnet-S affects categorization performance on them. **Top left:** Labeled Faces in the Wild (LFW) dataset. **Top right:** Contrast Sensitivity dataset. **Bottom:** Noisy Operators dataset. **B)** Top-1 test set accuracy of CORnet-S on the above datasets as a function of the fraction of synapses "injured", when introducing damage to V1, V2, V4, and IT. We plot the mean and $1\sigma$ over 5 runs with random seeds, capturing variability in the way a region is damaged every iteration.

## 3.2 Modeling Post-injury Plasticity

Next, we incorporate network plasticity during degeneration by retraining spared synaptic connections on real world visual stimuli that are chosen uniformly at random from ImageNet (section 4.2). We present each image once to the model to be retrained on between successive micro-events of a stroke—i.e., after every introduction of lesions. Furthermore, we try to understand the mechanisms underlying performance recovery due to plasticity through changes in neural predictivity and receptive fields (section 4.3). We compare how different layers of the models are able to predict real neuron responses in V1, V2, V4, and IT (Freeman et al., 2013; Majaj et al., 2015) of macaques with and without retraining during degeneration. Similarly, we visualize the receptive fields and plot the effective receptive field sizes of different convolutional layers of the model as follows (after Luo et al. (2016a)): Let $Y$ denote the output of the layer for which we want to plot the receptive field and $M$ the receptive field map (the input pixels). Since different layers receive different resolutions of input, we compute $Y_{\text{central}}$ to be the central part of $Y$ that has the same relative size across all layers. The receptive field is then given by the Jacobian:

$$\frac{\partial Y_{\text{central}}}{\partial M} \tag{1}$$

The locality of each receptive field map $M$ is summarized by computing:

$$\frac{1}{M_{\max}} \int_0^R r \mathbb{E}_r[M] \, dr \tag{2}$$

where $\mathbb{E}_r[M]$ is the expected value of the receptive field map $M$ at radius $r$.

## 4 Results

### 4.1 Localization of Focal Damage in the Network

We begin by analyzing how focal damage in different visual areas of the CORnet-S model impacts categorization performance across various visual tasks. Here, we did not model network plasticity, and it is incidental that we observe the following without the need to simulate recovery processes.

In figure 3B, we find that performance declines sharply when focal damage occurs in model visual areas that play a more significant functional role in a given task. For the Contrast Sensitivity dataset,

the task requires distinguishing luminance between the background and foreground while remaining color-invariant across contrast ratios, engaging layers in the "occipital lobe" such as model V1, V2, and V4 more than IT, which is positioned higher along the ventral pathway. This observation aligns with known biology, where early visual areas process low-level visual information, while IT contributes to more global recognition (Baker & Mareschal, 2001; Finn et al., 2007; Akbarinia & Gil-Rodriguez, 2020). In contrast, for the LFW dataset, we observe a more pronounced drop in performance when model V4 and IT are damaged. This outcome is biologically plausible due to the specialized role of the fusiform face area in higher-level face perception (Halgren et al., 1999; Kanwisher et al., 1997). Finally, on the Noisy Operators dataset, which requires extracting low-level edge features and forming a global understanding of shapes in noisy conditions, every visual area appears essential for task performance. Notably, the model maintains object recognition even when model V1 sustains up to 80% damage. Damage to model V1 and V2 results in a slower performance decline compared to model V4 and IT, showcasing the robustness of early visual areas in handling visual noise.

These findings suggest that carefully designed differential tasks can help localize focal damage to specific visual areas within the VVC. Such tests, like those used previously in a clinical study to localize agnosic visual disorders in humans with brain lesions (Martinaud et al., 2012), could serve as effective indicators of focal cortical damage, complementing the more reliable yet expensive Magnetic Resonance Imaging (MRI) and Computed Tomography (CT) brain scanning techniques.

## 4.2 CRITICAL THRESHOLDS FOR CATEGORIZATION PERFORMANCE RECOVERY

As a next step, we model post-injury plasticity during focal degeneration and evaluate object recognition performance on ImageNet. To make meaningful comparisons, we first apply global damage to various models of the VVC. In the absence of plasticity, categorization performance declines sharply as soon as the network sustains damage (figure 4A top). When plasticity is permitted, allowing the network to engage self-recovery mechanisms, the decline in performance is more gradual (figure 4A bottom).

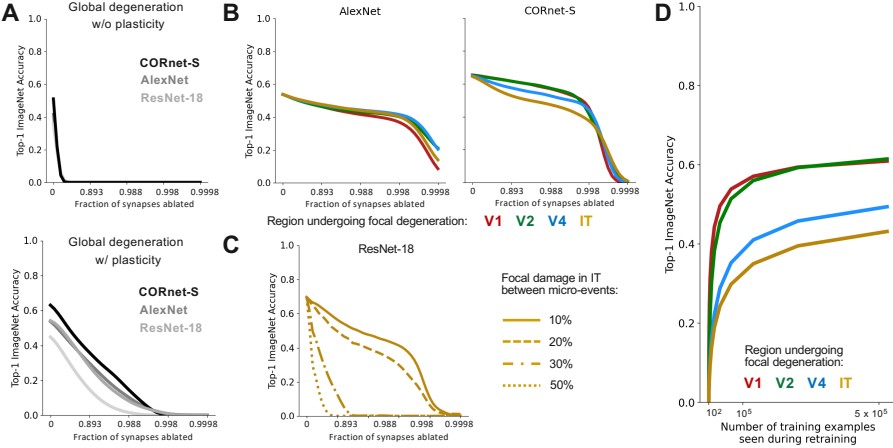

Figure 4: **Categorization performance after focal damage depends on critical thresholds, beyond which recovery becomes severely impaired. A)** ImageNet top-1 validation performance for different models under global network degeneration (**top**), along with the incorporation of plasticity (**bottom**). **B)** ImageNet top-1 validation performance for different models under focal degeneration, in the presence of plasticity. Damage level is set to 20% every lesioning iteration. Retraining occurs on a random subset of 500k images from ImageNet after every lesioning iteration. Each image is only presented once to the network during recovery. We take 500k images as a conservative amount by which recovery should have occurred in the network. Even if we assume that a patient sees a new image every 500ms, and that there is a new micro event (i.e., additional 20% damage) due to the ischemic stroke every 3 days, that is ∼518K images. **C)** ImageNet top-1 validation performance of ResNet-18 under focal damage to IT for different damage levels. **D)** ImageNet top-1 validation performance of CORnet-S as a function of the number of visual stimuli seen during recovery when V1, V2, V4, and IT are 95% damaged. All curves are smoothed using a 1D Gaussian filter with $\sigma$ = 2 for better interpretability.

In contrast to the effects observed during global damage, categorization performance following focal damage in the network depends on reaching a critical threshold. Initially, network performance declines gradually, but once this threshold is exceeded, a steep performance drop occurs (Figure 4B). The positioning of this critical threshold during the stroke's progression appears influenced by two key factors, among others: the specific model visual area sustaining damage (Figure 4B), and the extent of damage that accumulates between micro-events within the stroke (Figure 4C). Each visual area has distinct anatomical and functional characteristics, meaning their abilities to recover from damage vary. Additionally, differences arise from the degree of damage inflicted on synaptic connections within a visual area as the ischemic cascade unfolds. Notably, this threshold is not only apparent in categorization performance but also in the amount of diversity in real-world visual stimuli needed to be seen to recover a certain percentage, such as 70%, of lost visual function with respect to categorization (figure 4D). We observe that different images, on the order of $10^5$, are required between successive micro-events of a stroke for meaningful performance recovery. The specific amount, however, depends on the extent of focal damage sustained by the visual area in question.

Existing clinically-relevant literature on strokes indicates the challenges in diagnosing focal damage, often due to non-specific symptoms that fail to register on the National Institutes of Health Stroke Scale (NIHSS) (Martin-Schild et al., 2011). Many patients are unaware of their visual deficits, as described by Fisher (1986). Here, we computationally quantify these observations and make a prediction that post-injury plasticity aids in the recovery of object categorization performance up to a critical threshold, prior to which, symptoms may remain subtle or vague, making early detection difficult. Next we explore in more depth the mechanisms that underlie such recovery capabilities in different visual areas of the VVC.

## 4.3 Reorganization of Peri-lesional Neurons and Synapses for Recovery

Clinical research has shown evidence of functional and structural reorganization in spared neurons and synapses surrounding the lesions following focal damage (Liu et al., 2023). This reorganization can take place both within the damaged visual area or extend to nearby areas. In this experiment, we observe similar patterns of reorganization, validating the clinical findings. Importantly, the goal here is not to attribute specific recovery mechanisms to individual visual areas but rather to acknowledge the emergence of such processes across different visual areas in models of the VVC in a manner that mirrors biological observations.

In the previous experiment, we observed periods of slow deterioration of categorization performance before the critical threshold of recovery, precisely due to the presence of post-injury plasticity in the network. In the absence of neural plasticity, functional capacities in both the focal area undergoing degeneration and adjacent model visual areas decline sharply (figure 5A.1 left, A.3 top, B.1 top). Additionally, the receptive fields of neurons in these regions become increasingly ill-formed (figure 5A.2 top left), deviating from their typical Gaussian-like structure (Luo et al., 2016b). We further quantify this disruption by measuring the effective receptive field sizes, which degrade as damage accumulates (figure 5A.2 top right, B.2 top, B.3 top).

In the presence of neural plasticity, mechanisms of self-recovery manifest through functional and structural reorganization. Neurons in visual areas adjacent to the damaged region start assuming roles typically associated with the damaged area, becoming more functionally similar than they would in a healthy network state (figure 5A.1 middle and right, A.3 bottom). For example, when model V1 undergoes focal degeneration, its ability to predict real V1 responses in the VVC declines. However, neurons in model V2 begin compensating by better predicting real V1 responses, thus offsetting the loss of V1 function. This reorganization extends to the neurons' receptive fields, which, in the presence of plasticity, maintain their Gaussian-like structure. Furthermore, the effective receptive field sizes adapt to resemble those of the neighboring damaged visual area (figure 5A.2 bottom left and right). For instance, receptive fields generally increase as we progress from early to later layers of a neural network, a phenomenon well-established in neurobiology (Smith et al., 2001). One would believe that receptive fields are a characteristic entirely of the model architecture (Güçlü & Van Gerven, 2015), depending on properties such as the kernel size, stride, dilation, pooling operations, etc. Yet, in this case, the final convolutional layer of model V4, despite receiving the same input resolution, exhibits an increase in its effective receptive field under focal damage to model IT. This increase suggests that receptive fields are modulated not just by architectural parameters but by the distribution of kernel weights themselves—a consequence of post-injury plasticity in the network.

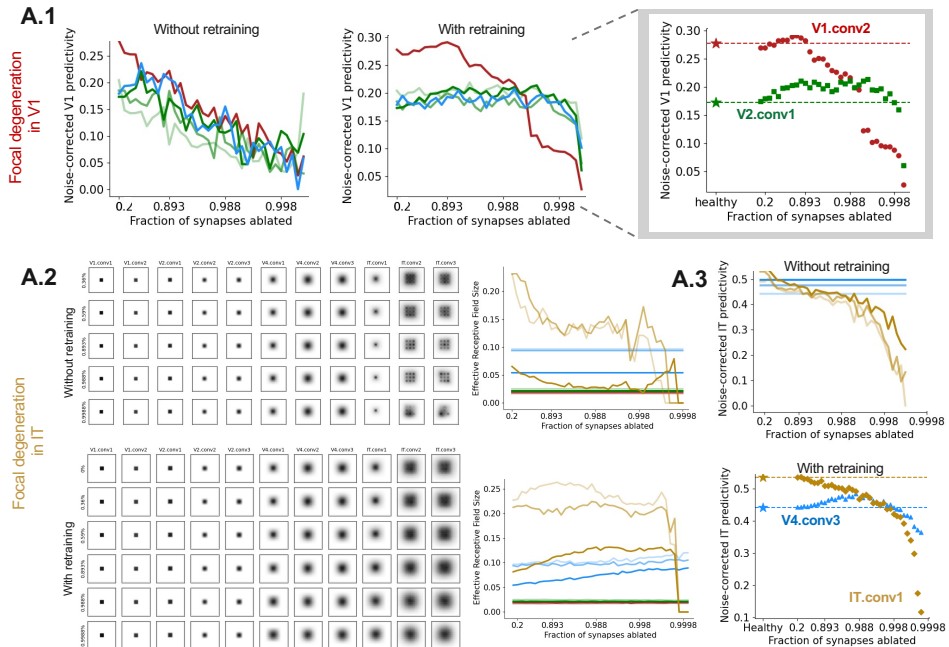

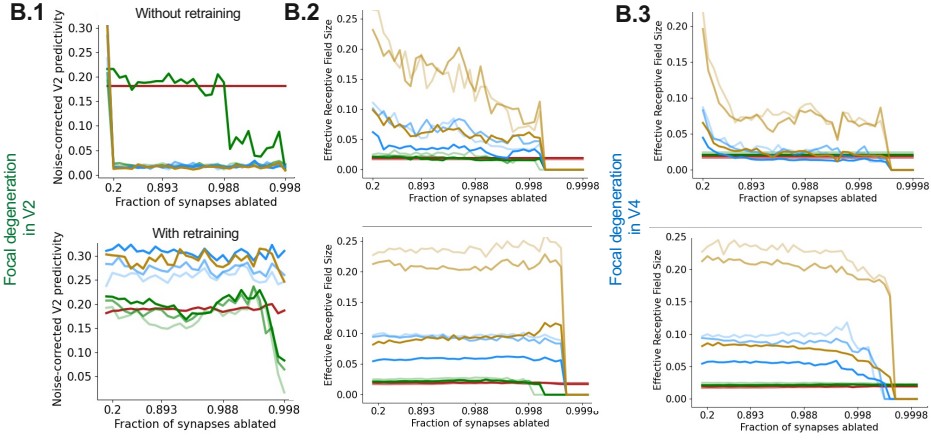

Figure 5: **Focal degeneration leads to the emergence of functional and structural reorganization of model neurons.** A set of plots that show two different mechanisms in CORnet-S that involve reorganization in **A)** regions at the borders of lesions, and **B)** spared synapses within the visual area incurring focal damage. Importantly, there was no difference in the underlying degeneration or plasticity scheme that led to the emergence of these two different reorganization mechanisms. **A.1**, **A.3**, **B.1** Noise-corrected predictivity scores—i.e., ability to predict real neuron responses in a specific visual area of primates—of different layers of the model with and without post-injury plasticity when focal damage occurs in different model visual areas. **A.2 left)** A visualization of the receptive fields and **A.2 right)**, **B.2)**, **B.3)** effective receptive field sizes of different convolutional layers of CORnet-S.

Recovery mechanisms are not confined to adjacent model visual areas alone. We also observe the capacity of spared peri-lesional synapses—i.e., those synaptic connections within the model visual area undergoing focal degeneration but not yet damaged (referred to as the penumbra)—to uphold both functional and structural characteristics of the damaged region. This occurs via plastic changes

in synaptic weights (figure 5B.1 bottom, B.2 bottom, B.3 bottom). Consequently, even though physiological and anatomical compensation from other model areas does not take place, the damaged model visual area retains its ability to predict real neuron responses and preserve the structure of its receptive fields despite ongoing degeneration.

## 4.4 ROLE OF SPARED RECURRENT CONNECTIONS IN RECOVERY

In the previous experiment, we noticed emergent reorganization in CORnet-S, a recurrent model. A consequent question to ask is if intra-layer recurrence played any role alongside neural plasticity to enable such reorganization. Having performed the same experiment in fully-feedforward networks AlexNet and CORnet-Z, we observe similar reorganization capabilities (figure 6A). This suggests that reorganization is mediated not by the presence of recurrent connections but by the ability of the brain to engage post-injury plasticity. Then, do intra-layer recurrent connections play any role in recovery of object categorization performance? We investigate this by removing them prior to introducing focal damage. Since recurrent connections do not add any trainable parameters to the network, any change in performance due to focal degeneration must be attributed to the importance of those connections in recovery. Our results indicate that spared recurrent connections play a significant role in facilitating quicker recovery (Figure 6B). When recurrent connections are injured, we observe an immediate decline in performance as damage is introduced. Notably, the performance curve consistently remains below that of a network with intact recurrent connections. As degeneration progresses, the performance gap becomes more pronounced, especially towards the tail end. This highlights a plausibly critical role of spared recurrent connections in modulating neuronal activity over time, allowing the network to adapt and compensate for synaptic loss in the affected area. However, a more detailed discussion begs a follow-up on the work.

## 4.5 ROLE OF TEACHING SIGNALS IN RECOVERY

So far, we have looked at how different biologically-plausible behaviors emerge in an artificial network following focal damage. The networks we used were trained through supervision, i.e., the cost function of the network was to optimize the learning of internal feature representations in the

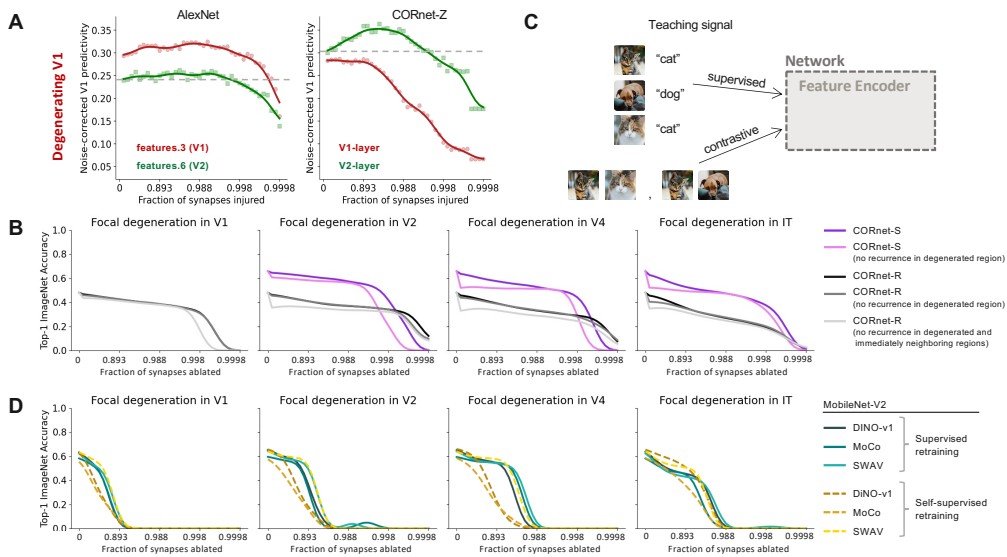

Figure 6: **Spared recurrent connections and providing teaching signals during the plasticity phase lead to improved performance recovery. A)** Noise-corrected predictivity scores of AlexNet and CORnet-Z under focal damage to model V1. Under focal degeneration to V1, V2, V4, and IT, we plot **B)** top-1 ImageNet accuracy of CORnet-S and CORnet-R with and without recurrent connections, **C)** schematic of different cost functions at play for a network, and **D)** top-1 ImageNet accuracy for MobileNet-V2, which is pre-trained according to different self-supervised objectives, when being re-trained on either those same objectives or through supervision during the plasticity phase. Note that CORnet-S does not have any recurrent connections in its V1. Curves in **A** are smoothed using 1D Gaussian filter with $\sigma = 2$ for better interpretability.

presence of a teaching signal—namely, category labels. However, cost functions in the brain are highly tunable, shaped by the animal's ethological needs (Marblestone et al., 2016). We then ask ourselves whether the presence of a teaching signal in the form of category labels, provided by the way in which a visual task is designed by a clinician for a patient during neurorehabilitation, leads to better recovery of object recognition performance after focal damage to a visual area. The two, of many, ways in which such visual tasks can be designed are by showing the patient (1) pairs of stimuli and their associated categories for scaffolded training, and (2) pairs of augmented versions of stimuli to pose a contrastive task in the absence of an explicit teaching signal.

We shift our focus from models pre-trained through supervision to models pre-trained using self-supervised contrastive learning techniques, which are more biologically plausible (Fodor & Crowther, 2002). Starting with MobileNet-V2 models pre-trained using these techniques, we introduce focal damage to different visual areas of the network, similar to previous experiments. To explore recovery mechanisms, we train the network using either the cross entropy cost function (as in earlier experiments) or continue optimizing it with the contrastive learning objective originally used during pre-training (figure 6C). We observe that the presence of a teaching signal in the form of category labels consistently leads to better categorization performance recovery than in their absence (figure 6D). This improvement is particularly notable in the early visual areas of the VVC hierarchy. Effective visual therapies in the form of scaffolded teaching for rehabilitation prove to be important in restoring to the patient their wholeness and quality of life (Suter & Harvey, 2011), and we hope that, over time, with the right model architecture, cost function, and learning rule, such therapies can be quickly tested as proof-of-concepts in-silico.

## 5 DISCUSSION

In this work, we presented a computational framework for modeling synaptic degeneration and post-injury plasticity in response to focal ischemic strokes to visual areas of the VVC. Our approach yielded novel insights into the mechanisms underlying recovery after focal brain injury due to synaptic dysfunction, generating testable predictions and validating existing clinical observations. Notably, we demonstrated that the functional signatures of well-designed differential tasks help in localizing focal damage, with these tests potentially serving as screening tools in clinical settings to aid in the early detection of neurodegenerative disorders. We also identified a critical threshold for categorization performance recovery, which may be particularly relevant for clinicians seeking to implement therapeutic interventions before significant deterioration in the network's visual capabilities occurs. Furthermore, we showed that neural plasticity, which facilitates recovery following focal damage, enhances categorization performance through the physiological and anatomical reorganization of spared neurons and synapses. Additionally, spared recurrent connections and scaffolded teaching signals during rehabilitation both contribute to improved performance recovery.

Importantly, these results are not meant for direct clinical translation at this stage; they serve to help us reverse engineer brain mechanisms to the extent that we can rely on the modeling assumptions made. This is an early work, and the hypotheses and predictions that we make here should be rigorously tested in vivo through neurophysiological experiments. We try our best to make as reliable predictions as we can by employing different model architectural classes and learning objectives. Although none of these models perfectly replicate the biological VVC, this does not preclude them from generating testable hypotheses (Golan et al. (2023); see also section 1). There are discrepancies between how we model focal damage and how stroke progression might occur in patients. However, modeling this spread in a way that generalizes across patients is challenging due to the high variability in stroke progression between patients (Salvalaggio et al., 2023). We model focal damage in a general way, such that the insights that we derive are still biologically-observed. Perhaps, future works can consider damage not just within a single visual area but also to synaptic connections between areas; for example, histopathologic changes due to an ischemic stroke are known to occur in nonischemic remote brain regions that have synaptic connections with the primary lesion site (Zhang et al., 2012). Additionally, having a model of the VVC that mirrors the underlying topography (Margalit et al., 2024) would enable more effective predictions. There is no model out there that implements both recurrence and topography, so it was a matter of what to focus on in this work.

We hope that future research will try to address these limitations to add to the contributions we make to the broader understanding of neural mechanisms underlying cortical damage and recovery.

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

# Appendix

## Table of Contents

## A  ADDITIONAL METHODS

### A.1  NEURAL NETWORK ARCHITECTURE AND TRAINING

In section 4.1, we fit a linear probe using the cross entropy loss to the output features from the penultimate layer of ImageNet pre-trained CORnet-S while keeping all weights from the base model frozen. For LFW alone, since the model is not specifically trained to work well with facial features, we first finetune the model parameters on a small subset of training data prior to linear probing. The linear probe is trained for 15 epochs with a batch size of 32. Parameters of the linear probe are optimized using Adam, with a learning rate initialization of 0.01 that decays according to a cosine learning schedule, and an $\ell_2$-regularization coefficient of 0.001. These hyperparameters were found by performing a stratified $k$-fold cross-validation procedure using the training set images, with $k = 5$ for the PICo MNIST and Noisy Operators dataset, and $k = 10$ for the LFW dataset.

In section 4.2, we incorporate post-injury plasticity by retraining all model weights. We start with different models pre-trained on ImageNet. Then, after every lesioning iteration, we retrain model weights using stochastic gradient descent with momentum ($\gamma = 0.9$). We initialize the learning rate to 0.001 and let it decay according to a cosine learning schedule. We employ a batch size of 128, an $\ell_2$-regularization coefficient of 0.001, and train the model for a single epoch. During this epoch, it sees $2^{19}$ images from ImageNet that are randomly chosen every lesioning iteration. We degenerate the visual areas for a total of $\lambda = 40$ lesioning iterations.

In section 4.5, we start with MobileNet-V2 that has been pre-trained on ImageNet using different self-supervised objectives, but with a view sampling mechanisms that optimize performance on this small-sized model (Tan et al., 2023). DINOv1 creates both global and several local views of a given image. All crops are passed to a student network, while only global views are given to the teacher network. It then trains the student network to match the output of the teacher network. MoCo performs contrastive learning between positive and negative pairs of image representations by maintaining a memory bank to store a large set of negative samples to compare with each positive example. SwAV contrasts different augmented views of the same image by assigning them to shared cluster prototypes. Similar to before, we only retrain the model on 500k images from ImageNet that are chosen uniformly at random for every lesioning iteration. We optimize model parameters by using the layer-wise adaptive rate scaling (LARS) optimizer (Ginsburg et al., 2018). We initialize the learning rate to 0.1, and then let it decay according to a cosine learning schedule with no warmup. We employ a batch size of 128 and an $\ell_2$-regularization coefficient of 1e-6. For supervised retraining, we use the same hyperparameters and optimizer as described in the previous paragraph.

All experiments are conducted on a single NVIDIA A40 GPU on an internal cluster.

### A.2  RECEPTIVE FIELD ANALYSIS

In section 4.3, we compute receptive fields for each model convolution layer using equation 1 over 1000 images chosen uniformly at random from the validation set of ImageNet. Receptive fields generated have a Gaussian distribution (see also (Luo et al., 2016a)) that are strongest at the center and then decay off toward the periphery.

### A.3  NEURAL PREDICTIVITY

In section 4.3, we compute the extent to which different convolution layers in CORnet-S can predict real neural responses in the brain through the Brain-Score platform (Schrimpf et al., 2018; 2020).

**Neural response dataset for V1 and V2.** (Freeman et al., 2013) generate the dataset by transforming samples of Gaussian noise to synthesize new images that have the statistical properties of 15 original photographs of visual textures. For each original texture, two sets of stimuli are generated using different statistics: 15 spectrally matched noise images and 15 naturalistic texture images (see (Freeman et al., 2013) for more details on stimuli generation). This dataset, comprising of a total of 450 unique images, are presented to 13 anesthetized macaque monkeys. Responses of 102 V1 and 103 V2 neurons are recorded to a sequence of texture stimuli, presented in suitably vignetted 4° patches centered on each neuron's receptive field. Each image was presented 20 times for 100ms, separated by 100ms of a blank gray screen.

**Neural response dataset for IT.** The dataset comprises of 2560 naturalistic stimuli from eight object categories (animals, boats, cars, chairs, faces, fruits, planes, and tables) (Majaj et al., 2015). 3D object models from these categories are pasted on naturalistic backgrounds after distorting their position, pose, and size. A circular mask is applied to each image (see (Majaj et al., 2015) for more details). These images are shown to two fixating macaques with two arrays places on the posterior-anterior axis of their IT cortices. The monkeys passively observe images for 100ms with a 100ms gap between successive images, each subtending approximately 8° visual angle. Sequences are repeated 50 times and recordings are taken from 168 IT neurons.

**Neural response fitting procedure.** According to (Schrimpf et al., 2018), source neuroids are mapped to each target neuroid using a linear transformation that is optimized using a partial least squares (PLS) regression with 25 components. Prior to performing this procedure, source features are projected into a lower-dimensional space using principal components analysis (PCA). 1000 principal components are retained from the feature responses per layer to 1000 ImageNet validation images that capture the most variance of a source model. This procedure is repeated for multiple train-test splits across stimuli. Predicted responses are compared with the measured responses by computing the Pearson correlation coefficient. The final predictivity score is the mean over across all train-test splits, with the predictivity score for each train-test split being the median computed over all individual neuroid neural predictivity values.

# B DATASETS

## B.1 CONTRAST SENSITIVITY DATASET

To capture the model's capability of processing low-level visual information about contrast under degeneration, we design the Contrast Sensitivity dataset—a slightly different version of the Pelli-Robson Contrast Sensitivity chart (Dg, 1988). The construction of the dataset involves assigning an image one of 12 background colors, picked to be evenly distributed over the color wheel. A digit $\in \{0, \ldots, 9\}$ is then superimposed onto the colored background, with the digit color derived by distorting the background color's RGB channels using randomly selected values.

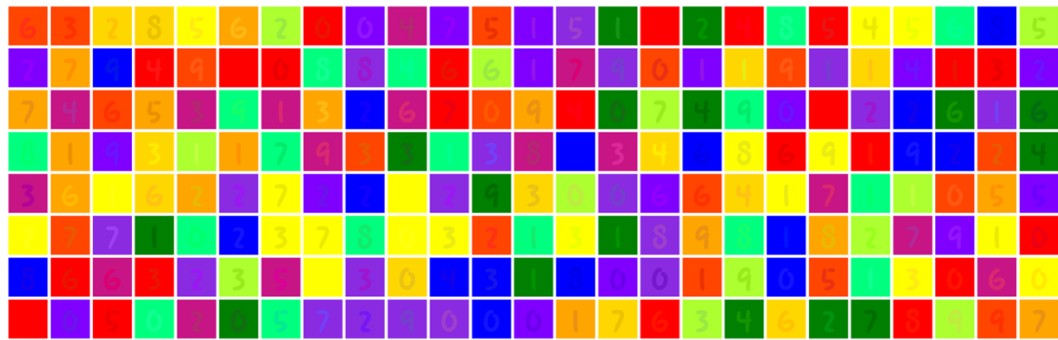

Figure 7: **Contrast Sensitivity dataset.** Example images chosen uniformly at random from the Contrast Sensitivity dataset.

---

**Algorithm 1** Pseudocode for computing the luminance of an RGB color

---

**Require:** `R` $\in [0..255]$: R value for the color
**Require:** `G` $\in [0..255]$: G value for the color
**Require:** `B` $\in [0..255]$: B value for the color
1: **for** `channel` $\in [$`R`, `G`, `B`$]$ **do**
2:    `channel` $\leftarrow$ `channel`$/255.0$
3:    **if** `channel` $\leq 0.03928$ **then**
4:       `channel` $\leftarrow$ `channel`$/12.92$
5:    **else**
6:       `channel` $\leftarrow ((\text{channel} + 0.055)/1.055)^{2.4}$
7:    **end if**
8: **end for**
9: `luminance` $\leftarrow$ `R` $* 0.2126 +$ `G` $* 0.7152 +$ `B` $* 0.0722$

---

To print a given digit $\in \{0, \ldots, 9\}$ onto an image with a given background color, lower and upper bounds for the contrast ratio, and whether the background should have a higher or lower luminance than the foreground, a $224 \times 224$ image is first constructed. A foregound color is found by continuously distorting the R, G, and B values of the background in either the positive or negative directions based on whether the foreground requires having a luminance greater or smaller than that of the background until the desired contrast ratio between the two colors is achieved. Both luminance and contrast ratio are evaluated according to the Web Content Accessibility Guidelines.[1] Algorithms 1 and 2 show more of the implementation details.

We allow the background color to take one of 12 possible values (which are uniformly spread over the color wheel):

- red (255, 0, 0)
- red-orange: (255, 69, 0)
- orange: (255, 165, 0)
- yellow-orange: (255, 215, 0)

---
[1]https://www.w3.org/TR/2008/REC-WCAG20-20081211/#contrast-ratiodef

---

**Algorithm 2** Pseudocode for generating an image from Contrast Sensitivity dataset of a given digit, background color, contrast ratio range, and luminance difference between the background and foreground colors.

---

**Require:** `digit` $\in [0..9]$: Number to be printed on the image
**Require:** `bg_color`: Background color in RGB format
**Require:** `contrast_ratio_low` $> 1$: Lower bound to the contrast ratio between the background and foreground colors as a floating point number
**Require:** `contrast_ratio_high` $>$ `contrast_ratio_low`: Upper bound to the contrast ratio between the background and foreground colors as a floating point number
**Require:** `luminance_diff`: $-1$ if luminance of the background should be more than the luminance of the foreground, and 1 if vice versa.
1: **for** $i \in$ `shuffle`$([0..255])$, $j \in$ `shuffle`$([0..255])$, $k \in$ `shuffle`$([0..255])$ in no particular order **do**
2:     `fg_color` $\leftarrow$ (`bg_color`$[0]$ + `luminance_diff` $* i$, `bg_color`$[1]$ + `luminance_diff` $* j$, `bg_color`$[2]$ + `luminance_diff` $* k$)
3:     `luminance_bg` $\leftarrow$ Luminance of the background color
4:     `luminance_fg` $\leftarrow$ Luminance of the foreground color
5:     `contrast_ratio` $\leftarrow$ (`L1` + 0.05)/(`L2` + 0.05), where `L1` is the relative luminance of the lighter of the colors, and `L2` that of the darker.
6:     **if** `contrast_ratio_low` $\leq$ `contrast_ratio` $\leq$ `contrast_ratio_high` **then**
7:       `image` $\leftarrow$ print `digit` in `fg_color` on a $224 \times 224$ image with a background color of `bg_color`
8:     **end if**
9: **end for**

---

- yellow: `(255, 255, 0)`
- yellow-green: `(173, 255, 47)`
- green: `(0, 128, 0)`
- blue-green: `(0, 255, 127)`
- blue: `(0, 0, 255)`
- blue-violet: `(138, 43, 226)`
- violet: `(127, 0, 255)`
- red-violet: `(199, 21, 133)`

The contrast ratios between the foreground and background colors lie in $[1.0, 1.5]$. We generate a total of 10000 training images (1000 images per digit) by randomly choosing a background color and then distorting the RGB values in either the positive or negative directions randomly by random amounts. To generate the test set, we perform the same procedure to generate a total of 1200 images (120 per digit; 10 images per background color; 5 images with a positive contrast ratio and 5 with a negative contrast ratio between the foreground and background colors). Sample images from this dataset are shown in figure 7.

### B.2 NOISY OPERATORS

We design the Noisy Operators dataset to evaluate a model's response to shape detection under noise. We do so by superimposing one of five binary operators $\in \{+, -, \times, /, \%\}$ in white on a black background and then introducing noise by randomly inverting (from black to white and vice versa) a given percentage $\in \{1, \ldots, 50\}$ of pixels.

Images from this dataset are generated in a similar way to that shown above. We start with an image with a black background and a white operator $\in \{+, -, \times, /, \%\}$ printed on it. We then add noise to the image based on a given percentage parameter $\mathcal{P} \in \{1, \ldots, 50\}$ by choosing $\mathcal{P}$ pixels uniformly at random and flipping their values from black to white and white to black. We generate a total of 6000 training images with the operator and $\mathcal{P}$ chosen uniformly at random (1200 images per operator), and a total of 1250 test images (250 images per operator; 5 images each with noise $\mathcal{P} \in \{1, \ldots, 50\}$). Sample images from this dataset are shown in figure 8.

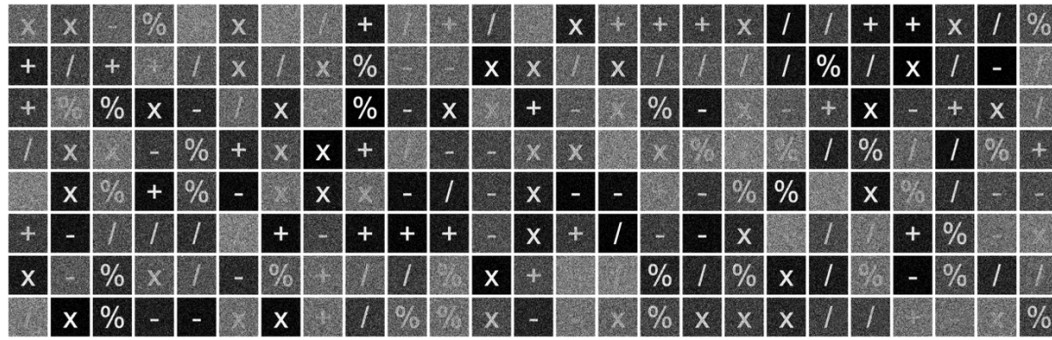

Figure 8: **Noisy Operators dataset.** Example images chosen uniformly at random from the Noisy Operators dataset.

### B.3 DATA RELEASE

We have created the above two datasets (Contrast Sensitivity and Noisy Operators) for understanding how different visual areas in the visual cortex respond to different tasks. To this end, we have created scripts that generate images that clinicians might use to detect visual deficits. None of the images contain any sensitive, confidential, or potentially derogatory information. The scripts are released as part of the code (under the MIT license). We also release the complete training and test images that we create from these scripts for reproducibility under the license CC BY 4.0.

### B.4 LABELED FACES IN THE WILD (LFW)

We use the LFW dataset (Huang et al., 2007b) with images aligned by funneling (Huang et al., 2007a) to pose a face verification task: given a pair of images, the model has to determine whether the two images belong to the same or different individuals.

To evaluate model performance on the LFW dataset, we follow the official protocol (Huang et al., 2007b) and previous works (Cox & Pinto, 2011; Bergstra et al., 2013) on generating train-test splits: we find the best hyperparameters based on 1000 images from "view 1" and perform evaluation by retraining the model with the best hyperparameters on 10 "view 2" splits of 6000 image pairs. The test set accuracy is the average over the accuracy values from these 10 splits. We use four element-wise comparison functions on the model features: product, absolute difference, squared difference, and square root of absolute difference. Because the model is not specialized to work with facial features, we finetune model parameters on 1000 image pairs once prior to introducing degeneration. This dataset is publicly available.

### B.5 IMAGENET

All models that we use are trained on ImageNet (Deng et al., 2009), and retraining after injury continues on ImageNet as well. ImageNet is a large-scale image dataset comprising of 1000 object categories, with 1281167 training images and 50000 validation images. It is available for free for non-commercial research.

# C Optimal stimuli analysis for V1 degeneration

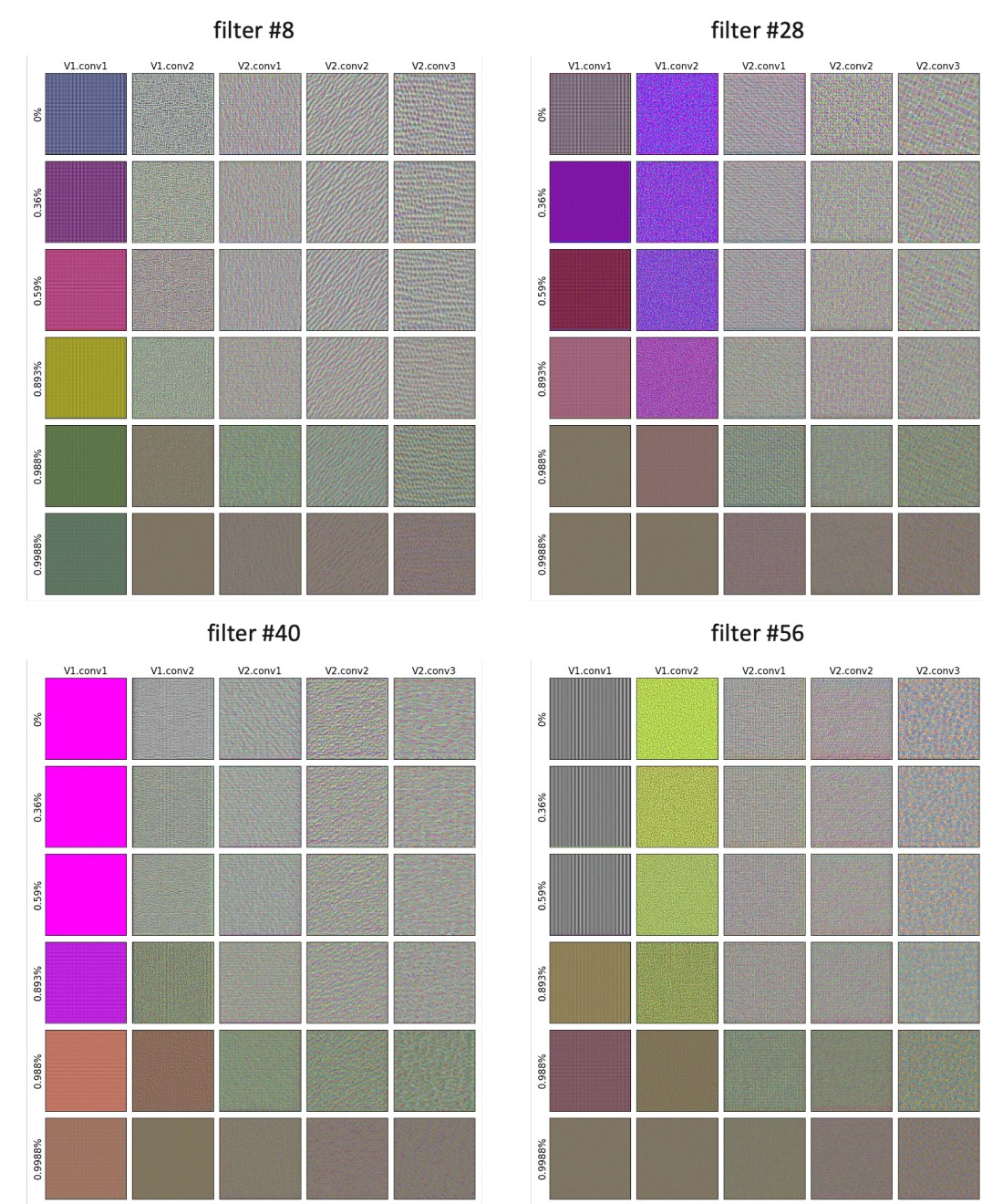

Figure 9: **Optimal stimuli analysis.** Optimal stimuli that most activate four different filters from different convolution layers of CORnet-S when degeneration with plasticity is introduced to V1.

During V1 degeneration, we notice a loss in orientation selectivity of first convolutional layer filters (section 4.3). What features do these filters then prefer to compute? We perform an optimal stimuli analysis[2] by starting with random noise and then iteratively trying to maximize the mean of the output of a certain filter from a certain layer of the model. The gradient is used to update the image so that, at the end of this procedure, we are left with an image that maximally activates that particular filter.

---

[2]https://github.com/utkuozbulak/pytorch-cnn-visualizations

We use an Adam optimizer with a learning rate of 0.1, $\ell_2$ regularization coefficient of 1e-6, and train it for 30 iterations.

By looking at figure 9, we find that the features that early visual areas V1 and V2 might be computing start decomposing into simpler-looking patterns with degeneration. There is a prominent drop in the ability to capture color, especially in the first convolution layer of V1. This might be because of synaptic damage to filters across different color channels. The sharpness of the feature being computed by these filters also fades with degeneration, possibly because of a reduction in the amount of information being received through spared synaptic connections. This is also evident from these layers' receptive fields starting to fade away, with the gaussian distributions that they might be resembling starting to flatten out (section 4.3). It is not entirely obvious if, with degeneration, V2 starts computing features that we as humans would visually associate as being what the real V1 prefers, even though we see the first convolution layer of V2 starting to better predict real V1 responses (section 4.3). One possible reason is that the input that V2 receives under degeneration is not the same as what a healthy V1 receives; the input is still affected by convolution and pooling layers in the damaged V1.

# D RECEPTIVE FIELDS FOR DIFFERENT MODEL LAYERS

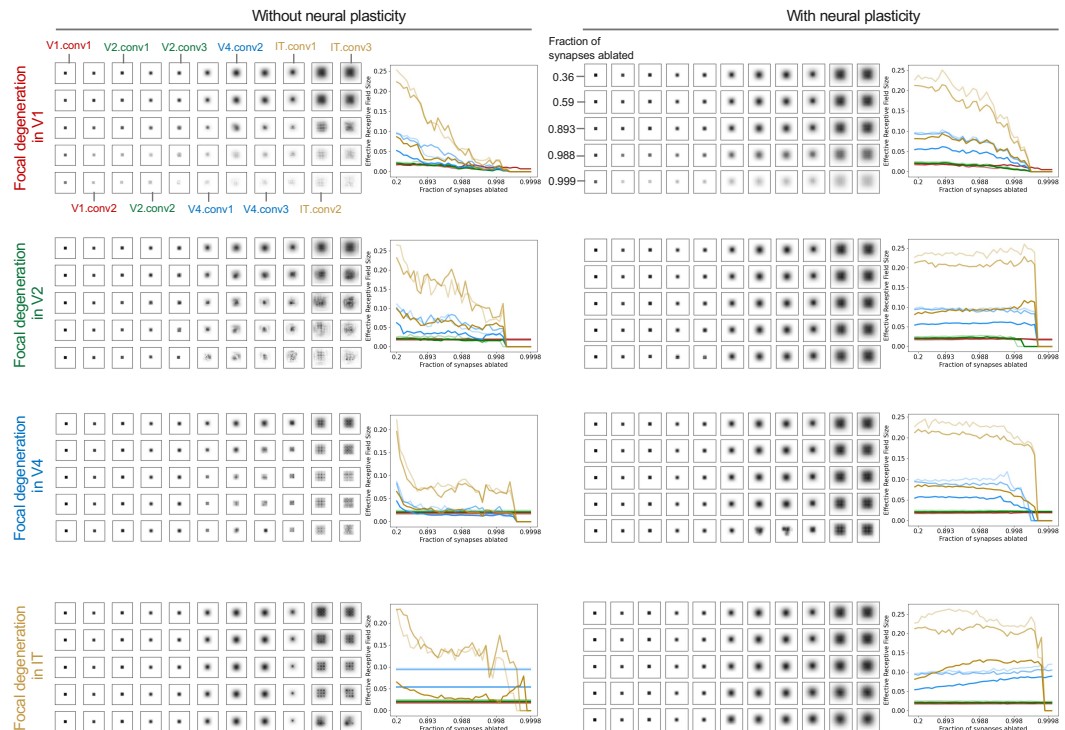

Figure 10: **Receptive fields under focal damage.** We visualize the receptive fields and effective receptive field sizes for different model layers of CORnet-S under focal damage to model V1, V2, V4, and IT.

## E CODE AND DATA AVAILABILITY

We share the datasets, and the code to generate these datasets, perform receptive field analysis, and model focal synaptic degeneration in PyTorch (Paszke et al., 2019) with this submission. All code is released under the MIT license, and all data under CC BY 4.0.

