# OpenReview forum: "Modeling Focal Synaptic Degeneration and Neural Plasticity in Ventral Visual Cortex"
_ICLR.cc/2025/Conference — Submitted to ICLR 2025_

### Official Review · Reviewer_iKAa · 2024-11-02

**Soundness:** 2
**Presentation:** 3
**Contribution:** 3
**Rating:** 6
**Confidence:** 4

**Summary:**

This study explores how synaptic degradation caused by stroke affects the ventral visual cortex (VVC) and assesses the role of neural plasticity in the recovery of visual functions post-injury. Using a primate visual cortex model, the study simulates synaptic degradation by progressively weakening synaptic connections in V1, V2, V4, and the inferior temporal cortex (IT), followed by retraining with real-world visual stimuli. The results indicate that synaptic degradation leads to a gradual decline in visual function within the VVC, followed by a sharp deterioration once a critical threshold is reached. Post-injury retraining reveals reorganization of surrounding neurons and enhanced recovery through preserved recurrent connections. Differentiated visual tasks also help pinpoint specific areas of damage within the VVC. Additionally, the study finds that retraining with contrastive learning protocols aids in the recovery of classification performance. Model simulations demonstrate that, in cases of localized synaptic degradation in the VVC, recurrent connections, types of visual stimuli, and the nature of training tasks significantly influence recovery outcomes.

**Strengths:**

This study presents significant innovations in understanding the impact of neural injury and synaptic degradation on ventral visual cortex (VVC) function. By simulating progressive weakening of synaptic connections, the research explores the effects of synaptic degradation on visual function, demonstrating the reorganization process of VVC neurons following synaptic injury and its contribution to functional recovery. This progressive synaptic degradation model offers a novel perspective, bringing originality to the field of neural plasticity research. Moreover, using a primate visual cortex model and real-world visual stimuli for experimental simulations, the study enables accurate tracking of visual function recovery. The retraining protocols and gradual degradation simulations are highly precise, uncovering the redistribution of neural load through recurrent connections in specific brain areas post-injury and demonstrating the feasibility of enhancing visual recovery through varied training tasks. Overall, this research holds considerable importance for neurorehabilitation and visual function recovery. The findings on the role of recurrent connections in network reorganization deepen our understanding of synaptic remodeling mechanisms after neural injury and provide a scientific basis for developing rehabilitation interventions centered on recurrent connections.

**Weaknesses:**

Although this paper presents a progressive synaptic degradation model and demonstrates its potential in visual cortex remodeling, the scope of experimental validation remains limited. Current experiments focus on specific visual tasks, without covering a broader range of visual functions or diverse damage models. Consequently, while the model performs well on the tasks tested, its generalizability and adaptability to other visual tasks remain uncertain. It is recommended to incorporate a wider variety of visual tasks, such as complex dynamic visual scenes or tasks requiring higher perceptual resolution, to comprehensively assess the model's practical effectiveness.

The model and experimental design also simplify biological complexity to some extent. For example, synaptic degradation is modeled as a gradual, linear process, but in actual neural injury, synaptic degradation is often nonlinear and accompanied by various physiological changes. This linear simplification may underestimate the complex factors present in real synaptic injury, potentially affecting the model's accuracy in practical applications. It is suggested to incorporate a more complex synaptic degradation mechanism or add a discussion explaining how these biological factors could be addressed in future research.

Additionally, while the paper demonstrates the role of recurrent connections in visual function recovery, the exploration of recurrent network reorganization mechanisms remains relatively superficial. The study does not delve into how specific network modules share or redistribute tasks post-injury, especially failing to clarify the roles of different network nodes in the functional reorganization process.

**Questions:**

In your model, you assume synaptic degradation as a gradual, linear process. However, in real neural injury, this process is typically more complex and nonlinear. Could you elaborate on how you address these simplified biological assumptions?

While the re-establishment of recurrent connections has shown effectiveness in visual recovery, we still lack a detailed understanding of the specific roles of individual modules within the network. Could you further explain or share insights on the roles of each module in the model, particularly in the processes of functional reorganization and recovery from injury?

Your study lacks comparisons with other baseline methods, such as non-recurrent network models or alternative multi-task learning approaches. Such comparisons would more clearly demonstrate the unique contributions of your model. Do you plan to conduct such comparative experiments, or could you clarify the reasons for not including these comparisons?

You mentioned the role of recurrent connections in the recovery process, but the paper does not delve deeply into the interaction between recurrence and neural plasticity mechanisms. Could you further elaborate on how these two aspects interact within the model?

---

> ### Author Response · Authors · 2024-11-25
> **Author Response to Reviewer iKAa (1/2)**
>
> We thank the reviewer for the discussion that they have brought up. We have tried to address all of them. Please also see the global rebuttal for common confusions around the intended purpose of the work. We reply to your concerns point-by-point below-
>
> ---
>
> > Current experiments focus on specific visual tasks, without covering a broader range of visual functions
>
> As stated in the paper, we model synaptic decay within the framework of the ventral visual cortex (VVC), which is implicated in core object recognition [1, 2]. Currently, ImageNet is the best-known dataset that when trained on makes the model effective in predicting responses of real neurons to a large degree [3]. Moreover, ImageNet is *not* a simple dataset, and almost every paper within the neuroscience community that focuses on the ventral visual pathway uses ImageNet for model training [4, 5, 6]. It is the current best known dataset that when trained on produces more neurally-aligned models of the ventral visual cortex. If a better dataset is developed, we would be happy to test our models on it when it becomes available.
>
> [1] DiCarlo, J. J., Zoccolan, D., & Rust, N. C. (2012). How does the brain solve visual object recognition?. Neuron, 73(3), 415-434.
>
> [2] Yamins, D. L., Hong, H., Cadieu, C. F., Solomon, E. A., Seibert, D., & DiCarlo, J. J. (2014). Performance-optimized hierarchical models predict neural responses in higher visual cortex. Proceedings of the national academy of sciences, 111(23), 8619-8624.
>
> [3] Yamins, D. L., & DiCarlo, J. J. (2016). Using goal-driven deep learning models to understand sensory cortex. Nature neuroscience, 19(3), 356-365.
>
> [4] Kubilius, J., Schrimpf, M., Kar, K., Rajalingham, R., Hong, H., Majaj, N., ... & DiCarlo, J. J. (2019). Brain-like object recognition with high-performing shallow recurrent ANNs. Advances in neural information processing systems, 32.
>
> [5] Margalit, E., Lee, H., Finzi, D., DiCarlo, J. J., Grill-Spector, K., & Yamins, D. L. (2024). A unifying framework for functional organization in early and higher ventral visual cortex. Neuron.
>
> [6] Dapello, J., Marques, T., Schrimpf, M., Geiger, F., Cox, D., & DiCarlo, J. J. (2020). Simulating a primary visual cortex at the front of CNNs improves robustness to image perturbations. Advances in Neural Information Processing Systems, 33, 13073-13087.
>
> ---
>
> > While the model performs well on the tasks tested, its generalizability and adaptability to other visual tasks remain uncertain
>
> We are not trying to generalize to other visual tasks. Please remember that we are working within the framework of the ventral visual cortex and not a vision world model. Could you elaborate on what “other visual tasks” we might have missed that are performed by the ventral visual cortex? Processing dynamic visual scenes or object motion usually happens in the dorsal visual cortex.
>
> ---
>
> > synaptic degradation is modeled as a gradual, linear process
>
> Synaptic degradation has not been modeled as a linear process. Degradation has been modeled as an asymptotic process where there is an exponential decay in the number of synaptic connections damaged with “time”. The progression of damage is highly variable between two real patients, meaning that we could never have modeled this process in a general way to understand brain function. Several previous works published in well-reputed neuroscience journals have used the same degradation protocol that we use here [1, 2, 3, 4], and we still observe biologically relevant behaviors in the model. We have provided the same argument in the discussion section of the updated paper.
>
> [1] Tuladhar, A., Moore, J. A., Ismail, Z., & Forkert, N. D. (2021). Modeling neurodegeneration in silico with deep learning. Frontiers in Neuroinformatics, 15, 748370.
>
> [2] Moore, J. A., Tuladhar, A., Ismail, Z., Mouches, P., Wilms, M., & Forkert, N. D. (2023). Dementia in convolutional neural networks: using deep learning models to simulate neurodegeneration of the visual system. Neuroinformatics, 21(1), 45-55.
>
> [3] Moore, J. A., Tuladhar, A., Ismail, Z., & Forkert, N. In Silico Modelling of Neurodegeneration Using Deep Convolutional Neural Networks. In SVRHM 2021 Workshop@ NeurIPS.
>
> [4] Moore, J. A., Wilms, M., Gutierrez, A., Ismail, Z., Fakhar, K., Hadaeghi, F., ... & Forkert, N. D. (2023). Simulation of neuroplasticity in a CNN-based in-silico model of neurodegeneration of the visual system. Frontiers in Computational Neuroscience, 17, 1274824.

---

> > ### Author Response · Authors · 2024-11-25
> > **Author Response to Reviewer iKAa (2/2)**
> >
> > > … affecting the model's accuracy in practical applications
> >
> > We never say in the paper that our aim is to deploy a (or any) model to a clinic. We are merely trying, as computational neuroscientists, to understand brain function under focal synaptic damage to the VVC. While hypotheses can be generated for rehabilitation using these models (as hinted in the discussion section), that requires thorough validation in animals first. We are not monkey electrophysiologists. But the fact that we try to make progress in this field through such “mechanistic” as opposed to “data-driven” models should speak to the scientific method of generating hypotheses, testing them, and then either accepting or rejecting them with a better version of the work. We are committed to the cause, and we plan to follow up.
> >
> > ---
> >
> > > How do recurrence and neural plasticity mechanisms interact with each other
> >
> > We explore the role of recurrence in recovery of object recognition performance in sections 4.3 and 4.4. From the models tested (both recurrent and purely feedforward), intra-layer recurrent connections do not seem to play a role in functional compensation (as opposed to its role in quicker recovery of object recognition performance). However, a more thorough analysis of the underlying changes in visual areas post recovery due to the presence of recurrent connections is beyond the scope of the current work.
> >
> > ---
> >
> > > Comparison with non-recurrent baselines
> >
> > We include a comparison with non-recurrent baselines (specifically, AlexNet and CORnet-Z) in section 4.4 of the paper.

---

### Official Review · Reviewer_AAV3 · 2024-11-03

**Soundness:** 2
**Presentation:** 2
**Contribution:** 2
**Rating:** 5
**Confidence:** 4

**Summary:**

This paper aims to study the process of focal synaptic degeneration in order to better understand the breakdown and possible recovery of biological neural network function in stroke. The authors use systematic weight pruning and network retraining experiments to examine how the deterioration of different components of multiple deep artificial neural networks leads to decrements in function on multiple behavioral tasks, as well as in similarity to neural data. The authors conclude the following:
- retraining allows networks to better maintain function in the presence of gradual degeneration of connections
- degeneration of different hierarchical levels leads to differences in the network's performance
- degeneration with retraining can lead to functional reorganization, whereby preserved part of the network can help to compensate for lost functions in degenerating parts of the network
- recurrent connections are critical for successful preservation of function
- self-supervised retraining is more effective than supervised retraining

**Strengths:**

- The paper is well written and tackles an important topic of both fundamental and applied interest
- The paper connects with neurology to a degree that is rare for visual cognitive computational neuroscience
- The authors use multiple datasets to test recovery
- The authors test multiple models
- The authors do not just test recovery performance, but also compare the match of unit properties to neural data in an areally mappable way, which is a huge strength that leads to discoveries of functional compensation
- The results on functional compensation are extremely interesting

**Weaknesses:**

- Figure 5 has some very interesting results regarding compensatory changes -- probably the most interesting results of the paper -- but the presentation of the results is very chaotic. It's unclear to me what the difference in analysis approach is in Figure 5B is vs. Figure 5A. Is it just the same approach for area V2? There are randomly some results from AlexNet, some from CorNet-S, some from CORNet-R, etc. It all feels a bit chaotic. More importantly, it seems the results may be cherry picked, and it is unclear how general the trend of compensatory changes is. Can the authors do a more systematic analysis of compensatory changes to present in supplementary, and use this figure to highlight a few representative results? For example, it would be useful to quantify the RF size changes for all pairs of areas, given the first of the pair being the one lesioned. Rather than plotting over the full range, a summary statistic could be constructed for a more concise presentation capable of showing the pattern across the full span of possible effects.
- There is no supervised control for Figure 6B. Without this control, it is inappropriate to claim that self-supervised vs. supervised learning in the retraining task matters for recovery performance, since the match between initial pretraining and recovery task could be equally if not more important.
- Perhaps the authors are just following the approach of Hinton, but I find the idea of plotting performance against lesion iteration rather than total fraction of lesioned synapses a bit odd. This makes Figure 4D very hard to interpret, since it is mentioned that 95% of connections have been lesioned for that analysis, but one cannot easily connect that number with the other plots. Additionally, I do not understand how the magnitude of performance is reaching nearly 80% in Figure 4D, when the network's unlesioned performance is less than 65% in Figure 4B.
- The degeneration here is completely random within an area, in contrast to the topographic specificity of lesions in biology. This seems likely to influence the pattern of results a lot, where large deficits in narrower domains could be observed with much less damage in a topographically organized network, with more preservation of other domains, due to the greater precision in mapping between damage and local function. It could be interesting for the authors to conceptually match this sort of damage by damaging units in a way that preserves correlation: e.g. after selecting some initial set of units, units with greatest activity correlation to these units are lesioned next, etc.
- Similarly, given the advancements in topographic neural networks in recent years, it is a bit disappointing that these authors do not discuss them at all. Indeed, these analyses would be very well suited for topographic models, since focal damage can be modeled much more accurately.
- The choice to preserve recurrent connections seems odd - why would these be particularly spared after a stroke? It is interesting to explore the effect of this computationally at the end, but it is a bit concerning that all of the preceding main results rely on this questionable choice.

**Questions:**

- Biologically, why is it more sensible to prune connections (as done here) rather than entire units? If there is a clear answer, it would be great for the authors to state it clearly in the text.
- See weaknesses for other questions

---

> ### Author Response · Authors · 2024-11-25
> **Author Response to Reviewer AAV3**
>
> We thank the reviewer for the discussion that they have brought up. We have tried to address all of them. Please also see the global rebuttal for common confusions around the intended purpose of the work. We reply to your concerns point-by-point below-
>
> ---
>
> > Presentation of results in Figure 5 is chaotic
>
> Thank you for pointing this out. We have updated the figure and the associated section for better interpretability. None of the results we present in the paper are cherry-picked, and we present effective receptive field sizes and neural predictivity scores for all model layers and damaged regions in the supplementary now.
>
> ---
>
> > Section 4.5
>
> In the earlier version of paper, we had mis-analyzed results for the difference in object recognition performance recovery due to supervised versus self-supervised retraining protocols. More specifically, the legend had been erroneously swapped for the models. Rest assured, we have fixed that error in the updated version of the paper, and revised the section to reflect that change. We apologize for this.
>
> ---
>
> > I find the idea of plotting performance against lesion iteration rather than total fraction of lesioned synapses a bit odd.
>
> We have fixed the x-axis to address this. Thank you for the feedback.
>
> ---
>
> > I do not understand how the magnitude of performance is reaching nearly 80% in Figure 4D
>
> We incorrectly divided the performance by 100 instead of a batch size of 128, which has been fixed to reflect in the figure now. Sorry for that.
>
> ---
>
> > Using topographic neural networks for modeling synaptic dysfunction
>
> We agree that a topographic neural network provides for an interesting analysis. We now discuss this in the discussion section of the updated paper. We are the first, to the best of our knowledge, to mechanistically interpret DANNs under focal synaptic dysfunction. Consequently, this is an early work. There is no model out there that implements both topographic constraints and recurrent connections (intra-layer or long-range feedback). So, it was a matter of deciding what to focus on in this paper. We are committed to the cause, and we do plan on following up with this work. Nevertheless, we hope that the analyses we present here are meaningful to the computational neuroscience community, and to the neuroscience community at large.
>
> ---
>
> > The role of recurrent connections in functional compensation
>
> We analyze the role that recurrent connections play, if any, in functional compensation in the updated version of section 4.3 of the paper. A key question that we answer there is – do we see functional compensation due to the presence of intra-layer recurrent connections along with neural plasticity, or does neural plasticity alone drive such reorganization. We do so by comparing CORnet-S with intact recurrent connections against purely feedforward DANNs, which still show such reorganization behaviors with neural plasticity.
>
> ---
>
> > Biologically, why is it more sensible to prune connections (as done here) rather than entire units?
>
> That is a valid point. As discussed in the introduction, in this work we focus on synaptic dysfunction, one of the earliest consequences of focal damage to a region. Synapses are highly sensitive to changes in its environment caused by nutrient depletion (oxidative stress, excitotoxicity, etc.). Their damage is seen to occur well before cell death [1]. One cause of why this might be is due to the involvement of neurotransmitters like glutamate which are quickly impaired by metabolic deficits.
>
> Verma, M., Lizama, B.N. & Chu, C.T. Excitotoxicity, calcium and mitochondria: a triad in synaptic neurodegeneration. Transl Neurodegener 11, 3 (2022). https://doi.org/10.1186/s40035-021-00278-7

---

### Official Review · Reviewer_g3ui · 2024-11-03

**Soundness:** 3
**Presentation:** 3
**Contribution:** 2
**Rating:** 5
**Confidence:** 4

**Summary:**

The paper describes a computational approach to understand the effects of localized synaptic damage in the ventral visual cortex caused by ischemic strokes. The researchers use deep artificial neural networks to mimic primate visual areas (V1, V2, V4, IT) and investigate how gradually losing synaptic connections influences visual processing. They also explore how the brain's ability to rewire itself, known as neural plasticity, helps in recovering from this damage. The study examines how degeneration impacts various visual tasks, identifies critical points for recovery, looks at the role of remaining connections, and evaluates the use of contrastive learning for rehabilitation. The findings offer insights into how the brain adapts after injury and propose strategies to improve recovery following cortical damage.

**Strengths:**

The paper presents a new computational framework for simulating focal synaptic degeneration in the VVC, a relatively underexplored area compared to global degeneration models. The innovative approach involves progressively damaging synaptic connections in biologically realistic DANNs and incorporating neural plasticity. The methodology is robust, utilizing various model architectures (feedforward, recurrent, self-supervised) to enhance the generality of the findings. The experiments are thorough, assessing the effects on different tasks and underlying mechanisms. The paper is well-written, with clear explanations and well-designed figures that aid understanding. Custom-designed datasets tailored to test specific functions add to the clarity. The study offers insights into how focal damage in visual areas affects function and recovery, potentially informing future therapeutic strategies, and bridges computational modeling with biological observations to enhance our understanding of neural plasticity.

**Weaknesses:**

While the models in the study are inspired by biological systems, they have limitations in fully capturing the complexities of real neural networks. The differences between simulated synaptic degeneration and actual stroke progression in patients may limit direct applicability. The modeling process assumes random synapse pruning, which oversimplifies the actual patterns of synaptic loss, and does not fully account for the complex time dynamics and spread of damage seen in real ischemic strokes. The retraining protocols used in the study involve large numbers of images between lesion iterations, which may not reflect the visual exposure patients experience in reality. Although the study shows contrastive learning to be effective over supervised learning, its practical use in clinical contexts could be challenging. Additionally, the paper notes that its hypotheses and predictions need validation through neurophysiological experiments, as the findings are currently limited to computational models.

**Questions:**

How would results change if the model used more realistic synaptic loss patterns from ischemic strokes, such as targeting specific neurons or connections? Can the model simulate more detailed temporal dynamics to better match clinical stroke timelines? If retraining data were greatly reduced to reflect patients' limited visual exposure, how would recovery be affected? Since contrastive learning aids recovery in the model, how can this be applied to practical rehab protocols? Could specific training tasks mimic these principles? Can this framework be adapted to other sensory systems or cognitive functions affected by strokes?

---

> ### Author Response · Authors · 2024-11-25
> **Author Response to Reviewer g3ui**
>
> We thank the reviewer for the discussion that they have brought up. We have tried to address all of them. Please also see the global rebuttal for common confusions around the intended purpose of the work. We reply to your concerns point-by-point below-
>
> ---
>
> > The differences between simulated synaptic degeneration and actual stroke progression in patients may limit direct applicability.
>
> This work is *not* for releasing a specific model for clinical application. Please see the global rebuttal for the motivation behind the work.
>
> ---
>
> > The modeling process assumes random synapse pruning, which oversimplifies the actual patterns of synaptic loss
>
> Again, please see the global rebuttal for a discussion on this. The progression of damage is highly variable between two real patients, meaning that we could never have modeled this process in a general way to understand brain function. Several previous works published in well-reputed neuroscience journals have used the same degradation protocol that we use here [1, 2, 3, 4], and we still observe biologically relevant behaviors in the model. We have provided the same argument in the discussion section of the updated paper.
>
> [1] Tuladhar, A., Moore, J. A., Ismail, Z., & Forkert, N. D. (2021). Modeling neurodegeneration in silico with deep learning. Frontiers in Neuroinformatics, 15, 748370.
>
> [2] Moore, J. A., Tuladhar, A., Ismail, Z., Mouches, P., Wilms, M., & Forkert, N. D. (2023). Dementia in convolutional neural networks: using deep learning models to simulate neurodegeneration of the visual system. Neuroinformatics, 21(1), 45-55.
>
> [3] Moore, J. A., Tuladhar, A., Ismail, Z., & Forkert, N. In Silico Modelling of Neurodegeneration Using Deep Convolutional Neural Networks. In SVRHM 2021 Workshop@ NeurIPS.
>
> [4] Moore, J. A., Wilms, M., Gutierrez, A., Ismail, Z., Fakhar, K., Hadaeghi, F., ... & Forkert, N. D. (2023). Simulation of neuroplasticity in a CNN-based in-silico model of neurodegeneration of the visual system. Frontiers in Computational Neuroscience, 17, 1274824.
>
> ---
>
> > The retraining protocols used in the study involve large numbers of images between lesion iterations
>
> A discussion on the number of images used for retraining, and why we use that number, is presented in lines 317-323 of the paper.
>
> ---
>
> > Although the study shows contrastive learning to be effective over supervised learning, its practical use in clinical contexts could be challenging.
>
> Please note that in the earlier version of paper, we had mis-analyzed results for the difference in object recognition performance recovery due to supervised versus self-supervised retraining protocols. More specifically, the legend had been erroneously swapped for the models. Rest assured, we have fixed that error in the updated version of the paper, and revised the section to reflect that change. We apologize for this.
>
> ---
>
> > Need validation through neurophysiological experiments
>
> That is beyond the scope of the current paper. We make it very clear in both the introduction and the discussion sections that outside of emergent reorganization that has been clinically observed in both animals and humans, the hypotheses we make do not replace in-vivo testing. To confirm such would require us to run a large neurophysiology experiment. We are not macaque electrophysiologists, so we don’t have the competence or the resource to perform what would be one of the most difficult experiments in neuroscience. However, as previously stated, why we believe that our hypotheses are worth acknowledging is due to the emergence of certain behaviors that *have* been clinically seen before.
>
> ---
>
> > Can this framework be adapted to other sensory systems or cognitive functions affected by strokes?
>
> Yes, absolutely. Given any computation model (trained to perform audition or decision making, or predict object motion, for example), an analysis of the form we have conducted here around neural predictivity, task performance, and population receptive field sizes, after careful translation to the applied domain, can be performed.

---

### Official Review · Reviewer_8GAG · 2024-11-03

**Soundness:** 2
**Presentation:** 2
**Contribution:** 1
**Rating:** 3
**Confidence:** 5

**Summary:**

The paper investigates the impacts of synaptic degeneration and post-stroke plasticity in the ventral visual cortex (VVC) via deep neural networks. The authors simulate focal degeneration in visual regions (V1, V2, V4, IT), in different layers of a neural network and examine recovery mechanisms. The study claims that  A) such mechanisms can be simulated from deep neural networks and B) the model’s recovery behavior under degeneration mirrors some biological observations. Additionally, the paper suggests that certain retraining protocols (e.g., contrastive learning) yield better recovery performance than supervised approaches.

**Strengths:**

1. Strokes and lesions are debilitating human conditions. Modeling these processes in neural networks might be a promising strategy.
2. Generating new recovery protocols is a potentially interesting use case of models

**Weaknesses:**

1. The study has major logical flaws with respect to the motivation. Specifically, the choice of using current neural network models as models of ischemic stroke and recovery is not clear. (see expanded notes below)
2. There are several unsubstantiated claims on biological relevance. The study intends to draw parallels between biological systems (brains) and models, and in the process overstates the degree to which this match is meaningful. The connection between model behavior under degeneration and observed behavioral phenomena (on which there is a lot in the rehabilitation and stroke literature) is speculative and not validated.
3. The clinical relevance of the findings are unclear. The paper’s findings and discussion on therapeutic interventions in its current form is superficial and lacks specific recommendations for clinicians.
4. The paper also relies heavily on prior existing models (like CORnets etc) without providing clear suggestions for improvements. The analyses repurpose existing methods and there is conspicuous lack of new contributions/innovations in the modeling approach (the models or the lesioning methods).

**Questions:**

1. Existing literature demonstrates that deep neural networks (DNNs) serve as reasonable predictive models of the visual cortex across species, including humans, primates, and rodents. DNNs tend to be most predictive when responses are represented as a linear combination of multiple model units, as single units alone do not capture brain responses as effectively. This study’s reasoning, however, seems to hinge on an assumed direct correspondence between individual model units and brain regions, as it involves lesioning units and model weights, which may not accurately reflect biological structures or processes.
2. Even beyond the assumption of a direct correspondence between model units and neurons, it’s important to recognize that deep neural networks (DNNs) are primarily viewed as models of inference rather than mechanisms for experience-dependent learning and plasticity. Current neural networks lack essential biophysical mechanisms, such as spike-timing-dependent plasticity (STDP), that underpin the complex processes of adaptation and reorganization observed in the brain following injury. Consequently, the model's findings may not translate accurately to biological systems. Simulating lesions in a model is one thing; however, claiming that these simulated lesions are analogous to or predictive of actual brain lesions requires strong empirical validation (even with existing data in the literature, something that is notably absent in this study).
3. Lets consider Section 4.1, which attempts to link layer-specific lesions with task performance across three tasks: color discrimination, noise discrimination, and face recognition. The authors’ main conclusion is that simpler tasks are processed by lower-level visual regions, while more complex tasks are handled by higher-level regions. However, this finding significantly oversimplifies the complexity of the actual biological phenomena observed in humans. For example, color processing in the human brain is not as low-level; rather, color-biased regions are found in the high-level visual cortex, “IT-like” areas. Studies by Conway et al., Pennock et al., and Lafer-Sousa et al. have shown that these higher-level areas are implicated in conditions like achromatopsia, where color perception is lost, which is not what the model’s analyses would predict. Turning to face processing, the paper similarly misses key complexities. In primates, including humans, face-selective patches are not limited to cortical regions but have also been observed subcortically, as shown in studies by Johnson et al., Gabay et al., and Kosakowski et al. This highlights that the neural architecture for face processing is more distributed and intricate. For examples lesions even to the subcortical amygdala network or early visual regions (like OFA) can result in highly specific face recognition deficits (like prosopagnosia). To accurately claim that the model lesions mimic specific cortical impairments in biology, needs stronger alignment with empirical data and a more sophisticated treatment of the neural networks. The true strength of in silico models lies in their ability to uncover complex, nuanced phenomena that might be challenging to observe directly in biological systems. However, rather than leveraging this potential, this paper settles for making simplified claims that do not fully capture the depth of these biological processes (and may not even require neural networks).
4. In Section 4.2, the authors suggest that their modeling strategy could aid in diagnosing micro-stroke events and quantifying their effects. However, the data presented in this section does not convincingly support this claim. The results rely on a broad 1000-way ImageNet classification task, which lacks the specificity required for detecting subtle, localized effects. Furthermore, the “focal” degeneration is implemented via “non-selective” pruning of filter weights within an area, which differs significantly from focal degeneration as it occurs in the brain. This creates a clear disconnect between the claim and the results: modeling broad accuracy changes does not directly address the detection of subtle, specific effects arising from transient events like micro-strokes.
5. The interpretation of recurrent dynamics in Section 4.4 raises a critical question about the validity of the method used, specifically whether an effective negative control is even possible. In this study, recurrence is implemented through backpropagation through time (BPTT), which is a standard approach for training recurrent neural networks but differs significantly from how recurrence operates in biological systems. The issue with using BPTT is that it limits the possibility of a true negative control for recurrence. In BPTT, once the recurrent connections are removed, the model effectively becomes a feedforward network without any capacity for dynamic feedback during task performance. Thus, any observed decline in performance may stem from this static structure rather than a specific role of recurrence in facilitating recovery. In order to really show that recurrence helps, the authors need to demonstrate that the opposite result is even viable with these current neural networks.
6. I found Section 4.5 of the paper particularly confusing. The authors motivate this section by describing some common visual rehabilitation tasks. The manipulation however however is the specific model cost function (self-supervised versus supervised model training). They show that self-supervised model recover somewhat faster (though unclear how much training data, including augmentation needed etc, which are critical for efficiency). The whole thing is mischaracterized as a self-supervised “retraining protocol” which is highly misleading and confusing with the rehabilitation protocols. Note that the discussion talks about this section with respect to rehabilitation which this section does not really engage with.
7. Overclaims and confusing terminology: The authors should clearly distinguish between models and brains (for example V1-layer for models and V1 for brains). This makes the paper very difficult to real and distinguish between experiments on models and the brain. The paper also overclaims the results in several places. The effects of strokes in the human brain are highly complex and depend on several factors. Especially strokes to the visual cortex in adulthood (fully trained networks) are often remarkably resilient and do not recover (unlike the picture presented in this study from simulations on neural networks).
Overall, while the paper presents a promising idea, the current implementation and the interpretation of modeling results do not align closely enough with actual observed behavioral effects to show the validity of the modeling strategy.

---

> ### Author Response · Authors · 2024-11-25
> **Author Response to Reviewer 8GAG (1/2)**
>
> We thank the reviewer for the discussion that they have brought up. We have tried to address all of them. Please also see the global rebuttal for common confusions around the intended purpose of the work. We reply to your concerns point-by-point below-
>
> ---
>
> > DNNs tend to be most predictive when responses are represented as a linear combination of multiple model units, as single units alone do not capture brain responses as effectively. This study’s reasoning, however, seems to hinge on an assumed direct correspondence between individual model units and brain regions, as it involves lesioning units and model weights
>
> This is not true. We **never analyze behaviors demonstrated by the model on a per unit basis**. DANNs themselves compute responses to visual stimuli as successive application of a linear combination of units within a model layer, followed by a nonlinear activation function. When we introduce synaptic activity decay to the model, it is introduced within a (or multiple) model layer(s). Analyses that follow include object recognition performance, capacity to predict real neural responses in different primate visual areas, and effective receptive field sizes, none of which are conducted on individual model units.
> - Object recognition performance is computed by scoring the number of correctly predicted categories for each test stimulus from the features computed by the entire model. This is done through linear probing, which is a linear combination of all output model units and not a one-to-one map between single units and categories.
> - Neural predictivity scores are computed over intermediate model layer activations and their ability to predict neural responses in a target brain region. The metric is the median over all individual neural recording predictivity values, and has been extensively used by prior works in comparing models to the brain (namely, through BrainScore [1, 2]).
> - Receptive fields have again been computed at the population level (i.e., quantified for a model layer as opposed to individual model “neurons”).
>
> [1] Schrimpf, M., Kubilius, J., Hong, H., Majaj, N. J., Rajalingham, R., Issa, E. B., ... & DiCarlo, J. J. (2018). Brain-score: Which artificial neural network for object recognition is most brain-like?. BioRxiv, 407007.
>
> [2] Schrimpf, Martin, Jonas Kubilius, Michael J. Lee, N. Apurva Ratan Murty, Robert Ajemian, and James J. DiCarlo. "Integrative benchmarking to advance neurally mechanistic models of human intelligence." Neuron 108, no. 3 (2020): 413-423.
>
> ---
>
> > DNNs are primarily viewed as models of inference rather than mechanisms for experience-dependent learning and plasticity.
>
> DANNs are not just used for data-driven modeling. Several well-cited papers within the computational neuroscience community have established the use of DANNs in **designing and implementing in a real monkey brain optimal perturbations that produce stronger responses in various areas of the VVS than any previously known natural stimulus**. This is necessary for us, because when we are perturbing different areas of the model, as we have done in this work, we want the responses to be good enough that they can be optimized to drive the brain. For example, Bashivan et al. [1] used a deep network to synthesize novel “controller” images based on the model’s implicit knowledge of how the VVS works to find that the model was capable of selectively controlling an entire neural subpopulation in subjects. Similarly, Walker et al. [2] trained an end-to-end, deep-learning-based model to synthesize optimal stimuli, which, for the mouse V1, drove neuronal responses in vivo significantly better than control stimuli. Thirdly, Ponce et al. [3] used a pre-trained deep generative neural network and a genetic algorithm to allow neuronal responses to guide the evolution of synthetic images that elicited large responses in various visual areas of macaques.
>
> Since DANNs have been shown to be effective at predicting such responses, given that synaptic dysfunction is a natural in-brain perturbation, DANNs seem that they are likely to be effective at predicting those as well.
>
> [1] Bashivan et al. “Neural population control via deep image synthesis”. In Science, 2019.
>
> [2] Walker et al. “Inception loops discover what excites neurons most using deep predictive models”. In Nat Neurosci, 2019.
>
> [3] Ponce et al. "Evolving images for visual neurons using a deep generative network reveals coding principles and neuronal preferences." In Cell, 2019.

---

> > ### Author Response · Authors · 2024-11-25
> > **Author Response to Reviewer 8GAG (2/2)**
> >
> > > Consequently, the model's findings may not translate accurately to biological systems.
> >
> > Through our analyses, we have shown the emergence of biologically-observed recovery protocols in the form of functional and structural compensation (through neural predictivity and pRF sizes). Moreover, we have used models (and many different ones to ensure reproducibility) that implement intra-layer recurrent connections, are trained through self-supervision, and are good at predicting responses and image-by-image human behaviors in the ventral visual cortex. **If DANNs did not model at least some aspects of the biological VVC, then no one within the neuroscience community would be using them for any kind of work for understanding the brain**. But that is not the case.
> >
> > ---
> >
> > > Claiming that these simulated lesions are analogous to or predictive of actual brain lesions requires strong empirical validation
> >
> > Firstly, we never claim that. We make it very clear in both the introduction and the discussion sections that outside of emergent reorganization that has been clinically observed in both animals and humans, the hypotheses we make do not replace in-vivo testing. Secondly, to confirm such would require us to run a large neurophysiology experiment. We are not macaque electrophysiologists, so we don’t have the competence or the resource to perform what would be one of the most difficult experiments in neuroscience. However, as previously stated, why we believe that our hypotheses are worth acknowledging is due to the emergence of certain behaviors that *have* seen clinically seen before.
> >
> > ---
> >
> > > Section 4.1
> >
> > In section 4.1, all we do is answer the question - can we localize where synaptic dysfunction has occurred in the ventral visual cortex using differential visual tasks. We do not claim that these are the exact tasks (or set of visual stimuli) that should readily be adopted by clinicians.
> >
> > ---
> >
> > > color processing in the human brain is not as low-level
> >
> > The pseudo-isochromatic MNIST task is not a task for color discrimination but rather contrast discrimination, which is thought to occur in early visual cortex. We apologize for the confusion, and we have updated the paper to make it more clear.
> >
> > [1] Boynton, Geoffrey M., Jonathan B. Demb, Gary H. Glover, and David J. Heeger. "Neuronal basis of contrast discrimination." Vision research 39, no. 2 (1999): 257-269.
> >
> > [2] Geisler, Wilson S., and Duane G. Albrecht. "Visual cortex neurons in monkeys and cats: detection, discrimination, and identification." Visual neuroscience 14, no. 5 (1997): 897-919.
> >
> > ---
> >
> > > Face-selective patches are not limited to cortical regions but have also been observed subcortically
> >
> > The alignment of DANNs of the ventral visual cortex and subcortical regions has not yet been quantified effectively, and doing so is out of scope for this paper.
> >
> > ---
> >
> > > The results rely on a broad 1000-way ImageNet classification task
> >
> > The ImageNet dataset is a carefully-curated dataset of foveated images of objects, and is the current best known dataset that when trained on produces more neurally-aligned models of the ventral visual cortex. If a better dataset is developed, we would be happy to test our models on it when it becomes available.
> >
> > ---
> >
> > > role of recurrence in functional compensation
> >
> > We analyze the role that recurrent connections play, if any, in functional compensation in the updated version of section 4.3 of the paper. A key question that we answer there is – do we see functional compensation due to the presence of intra-layer recurrent connections along with neural plasticity, or does neural plasticity alone drive such reorganization. We do so by comparing CORnet-S with intact recurrent connections against purely feedforward DANNs, which still show such reorganization behaviors with neural plasticity.
> >
> > ---
> >
> > > section 4.5
> >
> > We always compare models that are trained on the same amount of images, be it supervised or self-supervised. Additionally, please note that in the earlier version of paper, we had mis-analyzed results for the difference in object recognition performance recovery due to supervised versus self-supervised retraining protocols. More specifically, the legend had been erroneously swapped for the models. Rest assured, we have fixed that error in the updated version of the paper, and revised the section to reflect that change. We apologize for this.
> >
> > ---
> >
> > > use V1-layer for models and V1 for brains
> >
> > Thank you for the feedback. We have incorporated this terminology change.

---

> > > ### Comment · Reviewer_8GAG · 2024-11-27
> > >
> > > > If DANNs did not model at least some aspects of the biological VVC, then no one within the neuroscience community would be using them for any kind of work for understanding the brain.
> > >
> > > No one is denying that ANNs capture some aspects of biological VVCs—this is precisely why they have been useful for understanding the brain. However, by the same reasoning, there are also aspects of biological VVCs that ANNs, including DANNs, may not fully model. As a reviewer I am only trying to evaluate the authors' proposal against the data presented in the study. Being skeptical about a DANN-based procedure (as in this work) does not undermining the value of DANNs in neuroscience research.
> > >
> > > > Secondly, to confirm such would require us to run a large neurophysiology experiment.
> > >
> > > I totally understand that conducting a large-scale neurophysiology study is beyond the scope of this work. However, there is a wealth of existing literature on this topic, and incorporating or recapitulating a known phenomenon from these studies would greatly strengthen the argument. For this work to make a significant impact, it needs a compelling bridge between the proposal and the broader body of evidence in the field. This could be achieved by replicating the same kinds of tests conducted on human participants with stroke on in-silico models of stroke.
> > >
> > > > Section 4.1
> > >
> > > I request the authors to be more direct in their stated claims. In their response they note:  "Can we localize where synaptic dysfunction has occurred in the ventral visual cortex using differential visual tasks". Do they mean VVC in models or human brains? How would we know if their in-silico modeling proposal is appropriate?
> > >
> > > > color-processing in the human brain
> > >
> > > Thanks for this clarification. This is important and helpful.
> > >
> > > Please note that a number of my original questions and concerns have remained unanswered by the reviewers. No additional evidence (even from existing literature) has been presented in support of the the validity of the DANN-stroke approach. I remain unconvinced about the validity of the approach at this point.

---

> > > > ### Author Response · Authors · 2024-11-27
> > > >
> > > > Thank you for the follow up.
> > > >
> > > > ---
> > > >
> > > > > Validity of the approach
> > > >
> > > > > Being skeptical about a DANN-based procedure
> > > >
> > > > > Predicting responses to new stimuli is importantly distinct from perturbing the brain
> > > >
> > > > We would like to point out that we are not the first in using DANNs to model perturbations in the brain and understand how various behaviors emerge. Hinton, Plaut, and Shallice, back in 1993, were one of the first to simulate brain damage through their paper “Simulating Brain Damage”. The premise of their work? “Adults with brain damage make some bizarre errors when reading words. If a network of simulated neurons is trained to read and then is damaged, it produces strikingly similar behavior”. And they use a neural network to do this. The reviewer earlier claimed through their response that “DNNs are primarily viewed as models of inference rather than mechanisms for experience-dependent learning and plasticity”. Yes, DNNs have been used extensively as models of inference. But can the reviewer point to literature that claims that DNNs cannot be used to model learning? Hinton’s 1992 paper “How Neural Networks Learn from Experience” leads with the argument “Networks of artificial neurons can learn to represent complicated information. Such neural networks may provide insights into the learning abilities of the human brain”. Importantly, we never say, and nor does Hinton, that neural networks mimic the learning abilities of the brain. Merely that they can be used to provide insights. And if the reviewer feels that that is not strong enough of a motivation, then DNNs would never have been used in modeling brain perturbations. But Hinton goes on to do just that the following year.
> > > >
> > > > In the 1993 paper, the authors “selectively removed connections between neurons to see how their behavior changes”. The same approach for modeling perturbations (i.e., simulating synaptic loss) has been used by several other published papers to understand brain behavior [1, 2, 3, 4, 5]. They have perturbed synaptic connections. They have perturbed model units. They have introduced plasticity through retraining of network weights. They have used BrainScore to quantify a “damaged” model’s ability to predict responses to a healthy brain. We have done the same, albeit with more careful investigations. Instead of simply perturbing the entire network, we introduce damage focally. We look at brain predictivity, receptive fields, perturb recurrent connections in addition to synapses, and analyze how teaching signals affect object categorization performance recovery. We never make claims about brain behavior in general, but behavior in the sense of core object recognition. At the end of the 1993 paper, the authors write why their approach to DNN-perturbation modeling (which is, in essence, the same as what we do here) is successful in understanding the brain: “Instead of verbally characterizing each component in a complex neural mechanism and relying on intuition to tell us how damage will affect its behavior, we simulate that mechanism, damage it and watch to see what happens”. And even with the protocol they used for introducing said damage was simply selectively removing connections, the emergent behaviors were still relevant.
> > > >
> > > > The reviewer additionally asks that we do not recapitulate any know phenomena from prior studies on focal brain damage. We are quite surprised by this statement, given that we dedicate a significant amount of space and investigation to emergent functional and structural reorganization via two different mechanisms in DANNs, both of which have been biologically observed in humans, cats, mice, and monkeys. We cite a number of different studies that have shown this in the second paragraph of the introduction. We believe that this is a significant emergence in DANNs. In this way, we have “replicated the same kinds of tests conducted on human participants with stroke”. There will be an infinite number of tests that vary across individuals with stroke given different initial conditions, damage progression, etc. We observe emergent reorganization, one of the most widely reproduced behaviors across not just individual humans but across species.

---

> > > > > ### Author Response · Authors · 2024-11-27
> > > > >
> > > > > Finally, we share your skepticism around using DANNs to model perturbations. We never claim that the way in which we model focal damage, or that has been used by prior works, even though they replicate behaviors in the biological brain, is precisely how focal damage unfolds in the brain. But is that not the crux behind the scientific method – that a research scientist makes incremental efforts towards a goal through hypotheses that are either validated or rejected? We agree that modeling assumptions we make must be relaxed in future iterations of the work. And that would yield even better hypotheses around the underlying mechanisms. “Deep neural networks are not a single hypothesis but a language for expressing computational hypotheses” [5]. If, in the future, a better modeling approach or computational mechanism is used, whether using DNNs or without, to refute any predictions we make in this work, that would make us happy. We would be even happier if we are the ones to do it. DNNs provide falsifiable hypotheses, and in the process, drive improvement. We hope that the reviewer recognizes that when evaluating the paper.
> > > > >
> > > > >
> > > > > [1] Tuladhar, A., Moore, J. A., Ismail, Z., & Forkert, N. D. (2021). Modeling neurodegeneration in silico with deep learning. Frontiers in Neuroinformatics, 15, 748370.
> > > > >
> > > > > [2] Moore, J. A., Tuladhar, A., Ismail, Z., Mouches, P., Wilms, M., & Forkert, N. D. (2023). Dementia in convolutional neural networks: using deep learning models to simulate neurodegeneration of the visual system. Neuroinformatics, 21(1), 45-55.
> > > > >
> > > > > [3] Moore, J. A., Tuladhar, A., Ismail, Z., & Forkert, N. In Silico Modelling of Neurodegeneration Using Deep Convolutional Neural Networks. In SVRHM 2021 Workshop@ NeurIPS.
> > > > >
> > > > > [4] Moore, J. A., Wilms, M., Gutierrez, A., Ismail, Z., Fakhar, K., Hadaeghi, F., ... & Forkert, N. D. (2023). Simulation of neuroplasticity in a CNN-based in-silico model of neurodegeneration of the visual system. Frontiers in Computational Neuroscience, 17, 1274824.
> > > > >
> > > > > [5] Golan, Tal, JohnMark Taylor, Heiko H. Schütt, Benjamin Peters, Rowan P. Sommers, Katja Seeliger, Adrien Doerig, et al. (2023). “Deep Neural Networks Are Not a Single Hypothesis but a Language for Expressing Computational Hypotheses.” PsyArXiv. March 21. doi:10.31234/osf.io/tr7gx.
> > > > >
> > > > > ---
> > > > >
> > > > > > What I do say is that the authors have created lesions (strokes) based on sampling of model units
> > > > >
> > > > > In the paper, we “injure’’ (prune) synaptic connections between model “neurons’’. Pruning occurs indiscriminately over the space of synaptic connections within a focal region of the model, so yes, we can think of it as each synaptic connection in that region having a uniform probability of being sampled for pruning. However, the way in which model units are mapped to neural data still use PLSregression. In fact, we run our model on BrainScore to get neural predictivity scores in the paper. We do not simply use unit averaged responses to predict brain activity. The same approach was used by previous papers, as cited above.

---

> ### Comment · Reviewer_8GAG · 2024-11-26
>
> > This is not true.
>
> I am surprised and truly perplexed by the authors' response! The question I raised was about the mapping between models units and neurons/voxels. All the studies cited by the authors have used PLSregression (as on BrainScore) which is a linear-mapping strategy. And this is **exactly** my point. We know linear mapping can be used to predict brain data in monkey and neuron brains. Unit averaged responses (or uniform weighting) does not do as well at brain prediction. I haven't suggested in my review that the authors have analyzed behavior on a per-voxel basis.  What I do say is that the authors have created lesions (strokes) based on sampling of model units. This is akin to uniform weighted sampling.
>
> > real monkey brain optimal perturbations that produce stronger responses
>
> This response also does not address my question/concern. I am well aware of the studies the authors reference. Notably all of these utilized neural networks as models of **inference** (prediction ability on new stimuli) not learning (my original point). Predicting responses to new stimuli is importantly distinct from perturbing the brain (stroke) as they invoke fundamentally different mechanisms.

---

### Author Response · Authors · 2024-11-25
**Motivating the work, and revising the paper**

We thank all reviewers for their feedback on our work and having taken the time to read our paper. We sincerely hope that the following discussion will be meaningful in answering the concerns you have brought up, especially given the impact that you think this paper can make.

---

**Motivation behind the work**

This paper revolves around the central question - *What can deep artificial neural networks tell us about recovery from in-brain perturbations due to focal damage on a neural level?* This is a massive undertaking, and so we try to make the question addressable: we use *task-optimized* deep artificial neural networks; we analyze the *ventral visual cortex*; we consider a specific form of focal damage, namely, *synaptic dysfunction* (more discussion in the introduction and discussion sections); and recovery in the form of *object recognition performance*. If this was not clear earlier, we have revised the paper to be more explicit.

One might ask why these choices were made. It is important to remember that research in this area of mechanistic modeling has been sparse. This means that a phenomenon as complex and variable as focal damage in a complex system such as the brain cannot be studied in a single paper. And is that not the crux behind the scientific method – that a research scientist makes incremental efforts towards a goal through hypotheses that are either validated or rejected? We agree that modeling assumptions we make must be relaxed in future iterations of the work. And that would yield even better hypotheses around the underlying mechanisms. But the way in which we introduce synaptic dysfunction (like how prior works have introduced similar modeling assumptions) is still capable of producing emergent behaviors that have been biologically-observed. Again, if this was not made clear in the paper earlier, we apologize. We have now addressed this discussion in more detail in the paper.

---

**Paper Revision**

We have updated the paper with relevant discussions brought up by the reviewers, and we thank them for that. We believe that that has strengthened our paper. We are committed to the cause, and we hope that the analyses we present will be of benefit to the neuroscience community at large.

---

### Meta-Review · Area_Chair_sHuh · 2024-12-17

**Metareview:**

This paper models synaptic degeneration, e.g. triggered by a stroke, using a series of convolutional deep neural network models intended to serve as models of the ventral visual stream in primates. The authors then model synaptic degeneration using a pruning procedure that is localized to specific layers. Following that, they examine their ability to localize the damge using behavioural tests, how the models recover with further training, and which components of the model are critical for recovery.

The strengths of this paper are that it is tackling an area of connection between deep network models of the brain and neurology that is underexplored. It also provides some nice, intuitive results for how a network would recover to such an injury. The weaknesses are that it often obfuscates the difference between examining the model and real brains (e.g. by referring to layers of the ANNs as V1, V4, etc.), and it doesn't actually provide solid means for relating the results to real biological data, behaviors seen in stroke patients, or tests used by neurologists. As such, the insights provided are somewhat limited. As one reviewer noted in their reply to the authors, "The paper does not deliver on its stated goals of delivering actionable clinical insights into neurodegeneration." Therefore, a decision to reject was reached.

**Additional Comments On Reviewer Discussion:**

The reviewers raised a number of concerns, some of which are highlighted in the weaknesses the AC described. The authors attempted to provide a rebuttal to these concerns. However, as one reviewer noted in discussion, the rebuttals were generally combatative, and didn't attempt to address the concerns raised, so much as dismiss them. As such, the scores didn't really budge, and the final score was 4.75 on average, which is below the acceptance threshold.

---

### Decision · Program_Chairs · 2025-01-22

Reject